# L-CiteEval: Do Long-Context Models Truly Leverage Context for Responding?

## Abstract

Long-context models (LCMs) have made remarkable strides in recent years, offering users great convenience for handling tasks that involve long context, such as document summarization. As the community increasingly prioritizes the faithfulness of generated results, merely ensuring the accuracy of LCM outputs is insufficient, as it is quite challenging for humans to verify the results from the extremely lengthy context. Yet, although some efforts have been made to assess whether LCMs respond truly based on the context, these works either are limited to specific tasks or heavily rely on external evaluation resources like GPT-4. In this work, we introduce *L-CiteEval*, a comprehensive multi-task benchmark for long-context understanding with citations, aiming to evaluate both the understanding capability and faithfulness of LCMs. L-CiteEval covers 11 tasks from diverse domains, spanning context lengths from 8K to 48K, and provides a fully automated evaluation suite. Through testing with 11 cutting-edge closed-source and open-source LCMs, we find that although these models show minor differences in their generated results, open-source models substantially trail behind their closed-source counterparts in terms of citation accuracy and recall. This suggests that current open-source LCMs are prone to responding based on their inherent knowledge rather than the given context, posing a significant risk to the user experience in practical applications. We also evaluate the RAG approach and observe that RAG can significantly improve the faithfulness of LCMs, albeit with a slight decrease in the generation quality. Furthermore, we discover a correlation between the attention mechanisms of LCMs and the citation generation process.

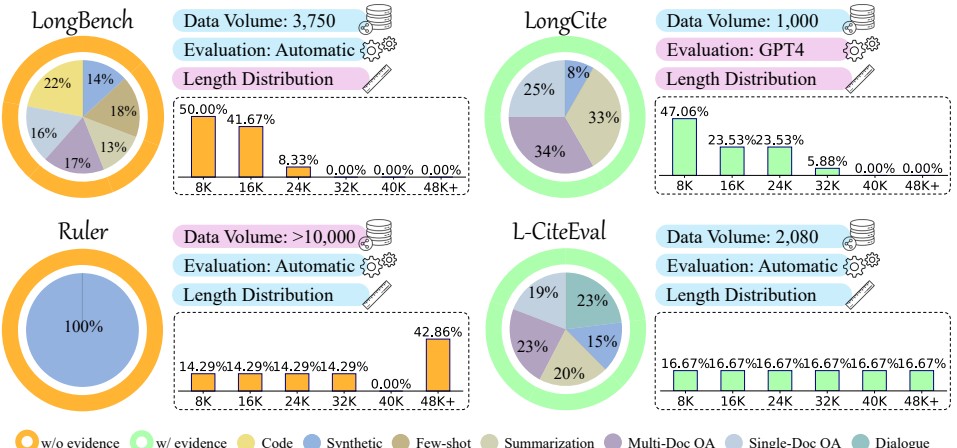

Figure 1: Overview and comparison among different representative benchmarks for LCMs.

## 1 Introduction

The rapid development of Long-context Models (LCMs) provides users with numerous conveniences in resolving long-context real-world tasks, such as code analysis (Zhu et al., 2024) and long document summarization (Reid et al., 2024). Recently, the community has gradually intensified its efforts to enhance the faithfulness of generative artificial intelligence (Manna & Sett, 2024),

which is of paramount importance for LCMs. This is because tasks that involve long context usually require LCMs to respond based on the provided context rather than relying solely on models' intrinsic knowledge. Therefore, there is an urgent need for a benchmark to verify whether LCMs truly leverage context for responding and reflect those models' capability on long-context tasks.

To date, substantial efforts have been made to develop benchmarks for evaluating LCMs. These endeavors aim to achieve several key objectives: (1) ensuring that the benchmarks include a **comprehensive** range of task scenarios and varying context lengths; (2) employing automated metrics to guarantee the **reproducibility** of evaluations; (3) incorporating an appropriate volume of test data to maintain evaluation **efficiency**; and (4) offering sufficient **interpretability** (e.g., providing evidence to support the responses). As shown in Fig. 1, taking three representative long-context benchmarks as examples: LongBench (Bai et al., 2023) primarily evaluates the accuracy of LCMs' responses across a range of realistic and synthetic tasks, with a context length of up to 24K tokens; Ruler (Hsieh et al., 2024) focuses on using synthetic data to test LCMs' capabilities in information retrieval over long sequences, with context lengths exceeding 48K tokens; and LongCite (Bai et al., 2024) assesses whether models respond based on the content within the context, employing GPT-4 as a judge. These benchmarks, based on their purpose, can be roughly divided into two categories: (1) evaluating long-context understanding and (2) assessing model faithfulness. The former evaluates model outputs using large volumes of test data to infer LCMs' capabilities but lacks interpretability to the generated results. The latter are mainly based on short-context datasets (e.g., in LongCite, the maximum sequence length only reaches 32K, comprising just 5.88% of the benchmark) and rely on external resources like GPT-4 to judge faithfulness, making the evaluation results hard to reproduce.

In this work, we introduce **L-CiteEval**, a comprehensive multi-task benchmark for long-context understanding with citations. As shown in Fig. 2, given the question and long reference context, L-CiteEval requires LCMs to generate both the statements and their supporting evidence (citations). There are **5** major task categories, **11** different long-context tasks, with context lengths ranging from **8K** to **48K** in L-CiteEval. To address the timeliness and the risk of data leakage in testing (Ni et al., 2024; Apicella et al., 2024), we incorporate 4 latest long-context tasks as the subsets in L-CiteEval, ensuring that the evaluation remains up-to-date and robust. Different from previous benchmarks for long-context understanding that primarily assess LCMs based on their predicted answers, L-CiteEval evaluates model performance based on both the generation quality (whether the predicted answer is correct) and citation quality (whether the provided citations can support the corresponding answer). To extend the context length of short-context data, we design a rigorous data construction pipeline to extend the sequence length and mitigate the perturbation introduced from the additional context. Additionally, to facilitate the ease of use and ensure reproducibility, L-CiteEval offers an automatic evaluation suite. Considering that the prediction from LCMs can be influenced by both the task difficulty and the context length, we propose two benchmark variants: **L-CiteEval-Length** and **L-CiteEval-Hardness**. These two variants strictly control the variables within the evaluation, focusing solely on context length and task difficulty to assess LCMs' capabilities.

We test 11 cutting-edge and widely-used LCMs, including 3 closed-source and 8 open-source models, which feature different sizes and architectures. We also explore whether the Retrieval-Augmented Generation (RAG) technique can improve the faithfulness of LCMs. Evaluation results indicate that there is a minor difference between open-source and closed-source models regarding generation quality, while open-source models substantially trail behind their closed-source counterparts in terms of citation quality. Utilizing the RAG technique exhibits a notable improvement in the faithfulness of open-source models, but it slightly impacts the generation quality. Furthermore, we reveal a correlation between the model's citation generation process and its attention mechanism (i.e., retrieval head (Wu et al., 2024)), demonstrating the validity of our benchmark and offering insights for future evaluations of LCM faithfulness and the development of advanced LCMs.

## 2 RELATED WORKS

### 2.1 LONG-CONTEXT UNDERSTANDING BENCHMARKS

Currently, there is a growing body of work dedicated to evaluating the long-context understanding capabilities of LCMs. The majority of benchmarks for LCMs are built based on real-world tasks that inherently encompass long context, including but not limited to long-document QA, long-document summarization, and long-term conversations (Li et al., 2023b; Shaham et al., 2023; An et al., 2023;

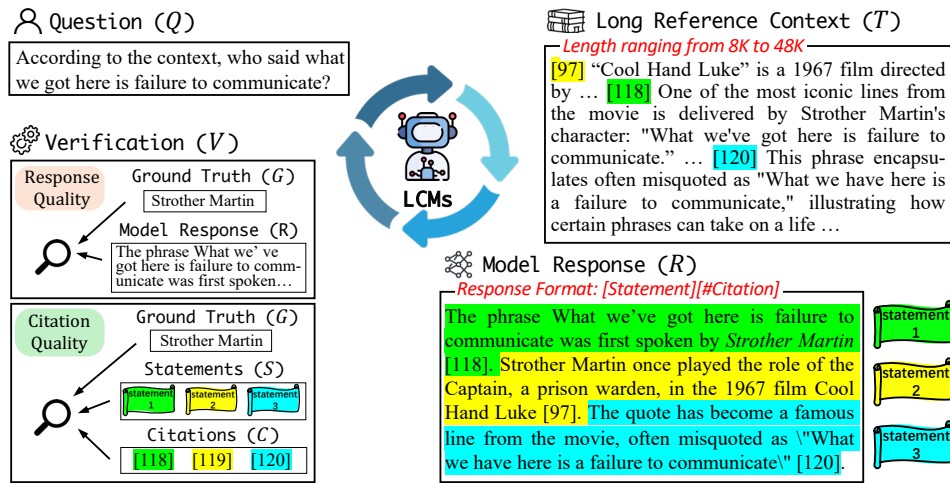

Figure 2: Task format and pipeline of L-CiteEval benchmark.

GoodAI, 2024; Bai et al., 2023; Dong et al., 2023; Zhang et al., 2024a; Lee et al., 2024; Levy et al., 2024). Recently, InfiniteBench (Zhang et al., 2024b) has pushed the boundaries of benchmarks based on real-world tasks by extending the context length beyond 100K tokens. However, real-world tasks exhibit a variety of forms and evaluation methods, and existing evaluations are applied inconsistently across different works. Additionally, the generated results can also be influenced by the intrinsic knowledge of LCMs. To make evaluations more controllable and eliminate the influence of the LCMs' intrinsic knowledge, synthetic benchmarks are often employed (Hsieh et al., 2024). Among those synthetic benchmarks, task formats can be custom-defined into various types, such as retrieval-based tasks that require the model to extract specific information from a long context (Kamradt, 2024; Mohtashami & Jaggi, 2023; Xiao et al., 2024; Liu et al., 2024; Wang et al., 2024), many-shot in-context learning (Agarwal et al., 2024; Bertsch et al., 2024), fact reasoning (Kuratov et al., 2024; Karpinska et al., 2024), *etc.* In this work, we introduce L-CiteEval, which contains both real-world tasks and synthetic tasks for long-context understanding with citations. By requiring LCMs to provide evidence to support their predictions, we can also mitigate the challenge of being unable to test whether LCMs respond based on their intrinsic knowledge or the provided context.

## 2.2 CITATION GENERATION

The citation generation task aims to verify whether the model predictions are supported by the referenced source (Li et al., 2023a). To evaluate the citations generated by models, Rashkin et al. (2023) first proposed *attributed to identified sources* (AIS) evaluation framework to measure the faithfulness of the model outputs. Then, some works began to improve the AIS framework in different tasks (such as single-document QA (Bohnet et al., 2022) and fact checking (Honovich et al., 2022)) and domains (such as science (Funkquist et al., 2022) and commerce (Liu et al., 2023)). To enhance the evaluation precision of citations within the generated text, Qian et al. (2023); Kamalloo et al. (2023); Li et al. (2023c) made great contributions based on the QA tasks. With the advancement of generative AI, citation generation has begun to require models themselves to generate citations that support their predictions (Gao et al., 2023). More recently, Bai et al. (2024) introduced *LongCite*, which represents the first attempt at citation generation in long context question-answering tasks. Compared with LongCite, L-CiteEval is (1) more comprehensive – it covers a wider range of tasks, supports longer context lengths, and strictly categorizes tasks by length intervals; (2) more reproducible – it relies entirely on automatic evaluation metrics without reliance on GPT-4 or human judgments; and (3) more efficient – the task and data distribution are well-designed in L-CiteEval, enabling users to utilize a limited amount of testing data to reflect the LCMs' capabilities.

## 3 L-CITEEVAL: TASK AND CONSTRUCTION

### 3.1 PROBLEM DEFINITION AND EVALUATION METRICS

**Problem Definition** As shown in Fig. 2, given the long context $T$ and question $Q$, a LCM is expected to generate the response $R$, which contains several statements $\mathcal{S} = \{s_1, s_2, \cdots, s_n\}$ and

Table 1: Statistic of tasks in L-CiteEval. The citation chunk size for each task is {*task*}:{*size*}.

| Tasks | Source | Evaluation Metric | Length Distribution | | | | | | Total |
|---|---|---|---|---|---|---|---|---|---|
| | | | 0~8k | 8~16k | 16~24k | 24~32k | 32~40k | 40~48k | |
| *Single-document QA* (NarrativeQA: 256, Natural Questions: 256) | | | | | | | | | |
| NarrativeQA | (Kočiskỳ et al., 2018) | Prec., Rec. | 40 | 40 | 40 | 40 | 40 | 40 | 240 |
| Natural Questions | (Kwiatkowski et al., 2019) | Prec., Rec. | - | - | 40 | 40 | 40 | 40 | 160 |
| *Multi-document QA* (HotpotQA: 128, 2WikiMultihopQA: 128) | | | | | | | | | |
| HotpotQA | (Yang et al., 2018) | Prec., Rec. | 40 | 40 | 40 | 40 | 40 | 40 | 240 |
| 2WikiMultihopQA | (Ho et al., 2020) | Prec., Rec. | 40 | 40 | 40 | 40 | 40 | 40 | 240 |
| *Summarization* (MultiNews: 128, GovReport: 128, QMSum: 128) | | | | | | | | | |
| MultiNews | (Ghalandari et al., 2020) | Rouge-L | 20 | 20 | 20 | 20 | 20 | - | 100 |
| GovReport | (Huang et al., 2021) | Rouge-L | 40 | 40 | 40 | 40 | 40 | 40 | 240 |
| QMSum | (Zhong et al., 2021) | Rouge-L | 20 | 20 | 20 | 20 | - | - | 80 |
| *Dialogue Understanding* (LoCoMo: 256, DialSim: 256) | | | | | | | | | |
| LoCoMo | (Maharana et al., 2024) | Prec., Rec. | 40 | 40 | 40 | 40 | 40 | 40 | 240 |
| DialSim | (Kim et al., 2024) | Prec., Rec. | 40 | 40 | 40 | 40 | 40 | 40 | 240 |
| *Synthetic Task* (NIAH: 256, Counting Stars: 128) | | | | | | | | | |
| NIAH | (Kamradt, 2024) | Rouge-1 | 20 | 20 | 20 | 20 | 20 | 20 | 120 |
| Counting Stars | (Song et al., 2024) | Accuracy | 30 | 30 | 30 | 30 | 30 | 30 | 180 |

their corresponding citations $\mathcal{C} = \{c_1, c_2, \cdots, c_n\}$. The context $T$ is divided into chunks of varying lengths based on the specific task, with each chunk representing a citation segment. Specifically, we set large citation chunk sizes for information-concentrated tasks like Single-Document QA to ensure segment integrity while using small citation chunk sizes for information-dispersed tasks like summarization to maximize the number of citations that LCMs can leverage to support the generated results. The model can then utilize these citation segments to support the statement $s_i$ within the response. In terms of output format, we require each statement $s_i$ to be strictly followed by a supporting citation chunk index $c_i$, which can also serve as an enclosure.

**Automatic Evaluation**   During the verification stage, the model response is evaluated from two aspects: the response quality and citation quality. As shown in Tab. 1, for response quality, we employ different evaluation metrics tailored to each specific task, e.g., Precision (Prec.) and Recall (Rec.) for QA tasks and Rouge-L (Lin, 2004) for summarization tasks. As for citation quality, following Gao et al. (2023), we adopt Citation Recall ($CR$) to reflect whether the model statements are fully supported by the citations; Citation Precision ($CP$) to detect irrelevant citations; and citation $F_1$ score to represent the overall citation performance. Besides, we report citation number $N$ to show how many citations the model uses to support its output. Different from previous works that utilize an NLI model (Gao et al., 2023) to automatically determine whether citations support the corresponding statements, we adopt a long-context NLI model deberta-base-long-nli (Sileo, 2024), to better align with long-context scenarios. We describe the calculation of $CR$ and $CP$ in Appendix B.

## 3.2 BENCHMARK CONSTRUCTION

There are 5 main categories in the L-CiteEval benchmark: Single-document QA, Multi-document QA, Summarization, Dialogue understanding, and Synthetic tasks, covering both realistic and synthetic tasks. We report the data source for each task in Table 1, For each task, we utilized the same construction process to handle the dataset. As shown in Fig. 3, the construction process for each task in the L-CiteEval benchmark consists of 3 steps, including (1) Seed Data & Padding Data Sampling, (2) Padding Data Filtering, and (3) Length Extension.

**Step1: Seed Data & Padding Data Sampling**   Considering the large amount of data in each source dataset, we first sample a portion of testing dataset $\mathcal{D}_{seed}$ as the seed data, from which we can subsequently construct the benchmark. However, some source datasets, e.g., LoCoMo (Maharana et al., 2024), exhibit short context. Consequently, we sample data from the remaining source dataset to serve as the candidate padding data $\mathcal{D}_{pad}$ for length extension. We divide all the sampled data ($\mathcal{D}_{seed}$ and $\mathcal{D}_{pad}$) into citation chunks of approximately equal size, with sentences as the basic unit. As mentioned above, we utilize different citation chunk sizes for different tasks. For tasks involving concentrated information, e.g., single-document QA, we employ smaller chunk sizes, while for tasks involving dispersed information, e.g., summarization, we use larger chunk sizes. We report the citation chunk size for each dataset in Table 1.

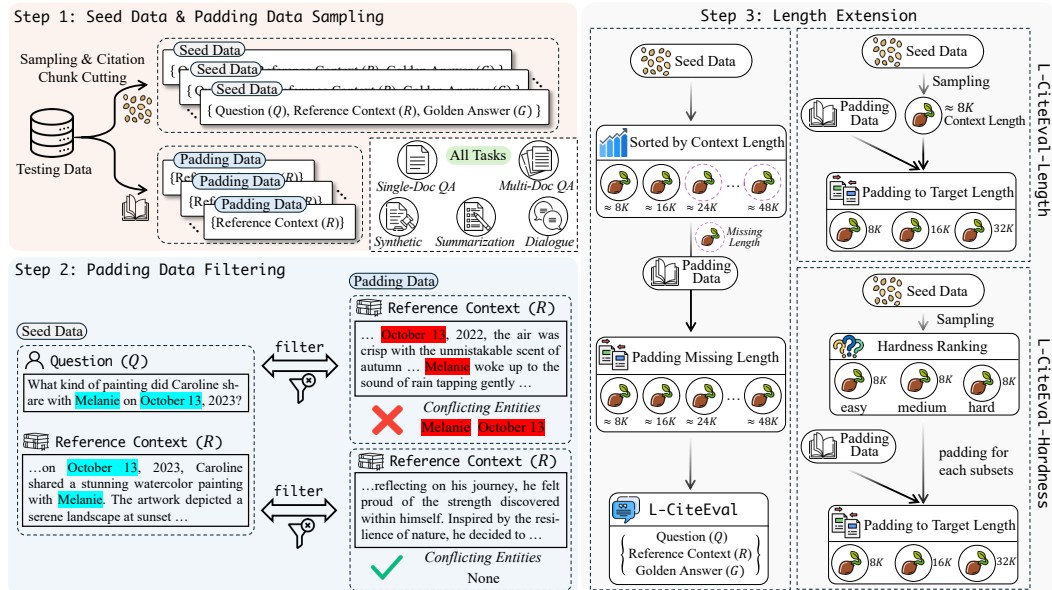

Figure 3: Benchmark construction pipeline.

**Step2: Padding Data Filtering** Using padding data to extend the length of the short-context dataset would introduce additional contextual information and could potentially influence the generated results. Therefore, we eliminate the padding data that might affect the predictions based on overlapping entities in the text. Specifically, we apply spaCy[1], a Named Entity Recognition model $f_\theta$, to extract all the entities $E$ from the question ($E_{seed}^{(Q)}$) and reference context ($E_{seed}^{(T)}$) in $\mathcal{D}_{seed}$, as well as the entities from the reference context ($E_{pad}^{(T)}$) in padding data. Then, we keep the padding samples $\mathcal{D}_{pad}^*$ that share a small entity overlaps with the seed data, which can be written as:

$$\mathcal{D}_{pad}^* = \left\{ \mathcal{D}_{pad}' \mid \mathcal{D}_{pad}' \sim \mathcal{D}_{pad}, \ E_{pad}^{(T)} = f_\theta(\mathcal{D}_{pad}'), \ |E_{seed}^{(T)} \cap E_{seed}^{(Q)} \cap E_{pad}^{(T)}| \le \delta \right\}, \quad (1)$$

where $\delta$ is the threshold to control the entity overlap between seed data and padding data. In this paper, we set this $\delta = 5$ as a strict criterion to filter out data that may potentially impact the results.

**Step3: Length Extension** After obtaining the padding data $\mathcal{D}_{pad}^*$, we leverage these data to extend the context length of seed data $\mathcal{D}_{seed}$. As shown in Figure 3, we have three different benchmark settings, including L-CiteEval and its two variants: L-CiteEval-Quality and L-CiteEval-Length. Specifically, for the L-CiteEval benchmark setting, given the target length interval of the dataset, we first sort the data according to the context length within each task. We then randomly sample contexts from $\mathcal{D}_{pad}^*$ to extend the context length and fill in the missing target length intervals. The L-CiteEval benchmark is designed to benchmark the models comprehensively. Thereby, the seed data and context extension data for all samples are different. For the L-CiteEval-Length benchmark, which aims to test the model's performance from the context length perspective, we use the same set of seed data and different sets of padding data to extend to various context lengths. For the L-CiteEval-Hardness benchmark that is designed to benchmark models based on question difficulty, we first quantify and rank the difficulty of each question according to the model's generation quality[2]. Then, we categorize the difficulty into three levels: easy (where the model mostly provides correct answers), medium, and hard (where the model mostly produces incorrect answers). We use the same padding data to extend the context length for each difficulty level. We use GPT-4 as the evaluator to classify the sample difficulty, as it shows the best generation quality.

**Benchmarks Overview** For clarity, we list the differences among the three benchmarks below:

---

[1] https://spacy.io/usage/models

[2] Specifically, we categorize the difficulty level of each sample based on GPT-4o because GPT-4o has been proven to exhibit the highest preference similarity with human annotators (Yadav et al., 2024).

Table 2: Statistic of LCMs. ∗ means the model utilizing YaRN (Peng et al., 2023) to extend the base context length. † denotes the MoE model, where activated parameters are enclosed in parentheses.

| Model | Ctx. Size | #Param | Architecture | Open-source |
|---|---|---|---|---|
| GPT-4o (20240513) (OpenAI, 2024a) | 128K | 🔒 | 🔒 | ✗ |
| o1-mini (OpenAI, 2024b) | 128K | 🔒 | 🔒 | ✗ |
| Claude-3.5-Sonnet (20240620) (anthropic, 2024) | 200K | 🔒 | 🔒 | ✗ |
| Qwen2.5-3B-Instruct (Team, 2024) | 32K (128K∗) | 3B | Decoder-Only | ✓ |
| Phi-3.5-mini-instruct (Abdin et al., 2024) | 128K | 3.8B | Decoder-Only | ✓ |
| Llama-3.1-8B-Instruct (Llama) | 128K | 8B | Decoder-Only | ✓ |
| GLM-4-9B-Chat (GLM et al., 2024) | 128K | 9B | Decoder-Only | ✓ |
| Mistral-NeMo-Instruct-2407 (Mistral, 2024) | 128K | 12B | Decoder-Only | ✓ |
| Qwen2-57B-A14B-Instruct (Yang et al., 2024) | 32K (128K∗) | 57B (14B†) | MoE | ✓ |
| Llama-3.1-70B-Instruct (Llama) | 128K | 70B | Decoder-Only | ✓ |
| Llama3-ChatQA-2-70B (Xu et al., 2024) | 128K | 70B | Decoder-Only | ✓ |

- **L-CiteEval** is designed to evaluate the comprehensive capabilities (generation quality and citation quality) of LCMs, which is constructed with different seed data (varying question difficulty) and padding data sources (varying context). This benchmark includes 2,080 testing samples, with 11 tasks across 5 categories.
- **L-CiteEval-Length** is designed to evaluate the LCMs from the context length perspective, which is constructed with the same seed data source (same question difficulty) but different padding data sources (varying context). This benchmark consists of 4 tasks across 4 categories, i.e., NarrativeQA (Single-Doc QA), HotpotQA (Multi-Doc QA), GovReport (Summarization), and Counting Stars (Synthetic task), with each task containing 200 testing samples. For each task, we establish three context length intervals: 8K, 16K, and 32K.
- **L-CiteEval-Hardness** is designed to evaluate the LCMs from the task difficulty perspective, which is constructed with the different seed data source (varying question difficulty) but same padding data sources (same context). This benchmark shares the same data distribution and volume with **L-CiteEval-Length**, except that the scoring is based on task difficulty (Easy, Medium, and Hard) rather than context length.

## 4 EXPERIMENTS

We conduct experiments with 11 latest LCMs, including 3 closed-source LCMs and 8 open-source LCMs, each with a context window size of at least 128K tokens, encompassing different parameters (ranging from 3B to 70B) and model architectures (dense model and MoE model). The statistic of LCMs is shown in Tab. 2. We provide one demonstration within the prompt for each task to make the model's output format more standard, i.e., one-shot learning during the inference time, and employ the same instruction for every LCM. Demonstration of model prediction, question, and instruction for each task is shown in Appendix F. We benchmark all the LCMs with **L-CiteEval** and then select 6 representative LCMs (including 1 closed-source LCMs and 4 open-source LCMs) to further evaluate on **L-CiteEval-Length** and **L-CiteEval-Hardness** benchmarks.

### 4.1 MODEL PERFORMANCE ON L-CITEEVAL

We report the citation quality in Tab. 3 ( information-concentrated tasks that require models to seek local information in several citation segments) and Tab. 4 (information-dispersed tasks that require models to seek global information from the entire context) and report the generation quality in Tab. 5.

### 4.1.1 ANALYSIS OF CITATION QUALITY

**Open-source LCMs versus Closed-source LCMs** Overall, there is still a significant performance gap between open-source LCMs and closed-source LCMs (excluding o1-mini), especially in tasks involving the reasoning step. Specifically, we can observe that: (1) closed-source LCMs generally provide more accurate citations (larger $F_1$ score) and tend to cite more segments with the context (larger value of $N$); (2) in the Dialogue Understanding task, the performance of the strongest open-source LCMs (Llama-3.1-70B-Instruct) has approached that of the closed-source LCMs. However, in other tasks requiring reasoning, particularly in synthetic tasks, although strong open-source

Table 3: Citation quality of LCMs in information-concentrated tasks within L-CiteEval.

| Models | Single-Doc QA | | | | Dialogue Understanding | | | | Needle in a Haystack | | | |
|---|---|---|---|---|---|---|---|---|---|---|---|---|
| | CP | CR | $F_1$ | N | CP | CR | $F_1$ | N | CP | CR | $F_1$ | N |
| 🔒 *Closed-source LCMs* | | | | | | | | | | | | |
| GPT-4o | 32.05 | **38.12** | 33.48 | 2.02 | 53.90 | **64.25** | **56.76** | 2.17 | **76.25** | **76.67** | **76.39** | 1.12 |
| Claude-3.5-sonnet | **38.70** | 37.79 | **37.43** | 3.54 | **54.45** | 50.48 | 51.45 | 2.83 | 65.00 | 68.33 | 65.97 | 1.04 |
| o1-mini | 29.83 | 35.33 | 31.66 | 3.38 | 45.54 | 50.74 | 47.21 | 2.63 | 25.42 | 28.33 | 26.25 | 1.58 |
| 🔓 *Open-source LCMs* | | | | | | | | | | | | |
| Qwen2.5-3b-Ins | 7.13 | 5.83 | 6.00 | 1.75 | 9.53 | 9.71 | 8.41 | 2.33 | 12.08 | 12.50 | 12.22 | 1.12 |
| Phi-3.5-mini-Ins | 21.06 | 20.46 | 19.14 | 2.86 | 20.39 | 24.27 | 20.57 | 2.27 | 11.11 | 12.50 | 11.53 | 1.20 |
| Llama-3.1-8B-Ins | 22.68 | 24.73 | 22.64 | 2.59 | 51.86 | 57.58 | 53.50 | 2.08 | 34.31 | 35.83 | 34.72 | 0.99 |
| Glm-4-9B-chat | **29.00** | **28.66** | **28.05** | 2.21 | **54.54** | 55.62 | **53.58** | 1.78 | 46.53 | 50.83 | 47.78 | 1.23 |
| Mistral-Nemo-Ins | 4.34 | 3.68 | 3.76 | 0.68 | 23.91 | 24.33 | 23.50 | 1.35 | 11.11 | 12.50 | 11.53 | 1.18 |
| Qwen2-57B-A14B-Ins | 4.90 | 3.43 | 3.82 | 1.27 | 22.63 | 22.54 | 21.61 | 1.80 | 15.28 | 15.83 | 15.42 | 1.17 |
| Llama-3.1-70B-Ins | 25.89 | 26.89 | 26.11 | 1.23 | 51.71 | 56.20 | 53.19 | 1.76 | **46.67** | 46.67 | 46.67 | 0.82 |
| ChatQA-2-70B | 21.75 | 22.54 | 21.92 | 1.12 | 47.67 | 51.25 | 48.77 | 1.29 | 38.33 | 38.33 | 38.33 | 0.95 |

Table 4: Citation quality of LCMs in information-dispersed tasks within L-CiteEval.

| Models | Multi-Doc QA | | | | Summarization | | | | Counting Stars | | | |
|---|---|---|---|---|---|---|---|---|---|---|---|---|
| | CP | CR | $F_1$ | N | CP | CR | $F_1$ | N | CP | CR | $F_1$ | N |
| 🔒 *Closed-source LCMs* | | | | | | | | | | | | |
| GPT-4o | 57.48 | **58.50** | 56.10 | 1.71 | 34.37 | 54.28 | 41.60 | 22.86 | **83.37** | **81.18** | **81.71** | 4.54 |
| Claude-3.5-sonnet | **66.85** | 55.62 | **58.58** | 2.44 | **36.70** | 55.03 | **43.45** | 17.70 | 73.01 | 75.83 | 73.15 | 4.81 |
| o1-mini | 49.95 | 49.60 | 48.58 | 1.78 | 20.23 | 33.61 | 24.83 | 19.58 | 34.06 | 46.46 | 38.45 | 6.73 |
| 🔓 *Open-source LCMs* | | | | | | | | | | | | |
| Qwen2.5-3b-Ins | 13.17 | 8.04 | 9.37 | 1.96 | 7.72 | 12.15 | 9.09 | 9.52 | 3.82 | 1.48 | 2.01 | 1.66 |
| Phi-3.5-mini-Ins | 11.89 | 10.25 | 10.53 | 1.71 | 10.90 | 10.94 | 9.60 | 8.23 | 4.19 | 3.67 | 4.09 | 3.48 |
| Llama-3.1-8B-Ins | 43.41 | 42.15 | 41.64 | 1.62 | 19.57 | 23.03 | 20.83 | 18.31 | 16.87 | 18.26 | 19.18 | 4.19 |
| Glm-4-9B-chat | 47.91 | 44.75 | 45.09 | 1.64 | **29.16** | **37.29** | **31.92** | 11.38 | 18.15 | 15.69 | 16.21 | 4.52 |
| Mistral-Nemo-Ins | 17.61 | 15.45 | 15.85 | 0.70 | 11.21 | 14.85 | 12.40 | 5.45 | 3.09 | 2.92 | 3.26 | 2.32 |
| Qwen2-57B-A14B-Ins | 17.30 | 12.07 | 13.61 | 1.06 | 4.01 | 3.37 | 3.19 | 3.81 | 4.37 | 4.37 | 4.24 | 4.24 |
| Llama-3.1-70B-Ins | **49.64** | **54.02** | **50.74** | 1.42 | 25.50 | 31.99 | 27.91 | 11.78 | **66.85** | **61.74** | **63.73** | 4.37 |
| ChatQA-2-70B | 47.20 | 49.51 | 47.92 | 1.10 | 19.57 | 23.60 | 20.89 | 11.81 | 14.02 | 11.22 | 13.22 | 3.49 |

LCMs like GLM-4-9B-Instruct cite a similar number of segments as the closed-source models, the quality of these citations is lower, resulting in a performance gap of nearly 20 $F_1$ points.

**Performance of Open-source LCMs** In general, there is significant room for open-source LCMs to improve, and medium-sized open-source LCMs (Llama-3.1-8B-instruct and GLM-4-9B-Chat) are highly competitive, with performance that matches or even exceeds that of large LCMs (Llama-3.1-70B-instruct). More concretely, our findings are: (1) The improvement in citation quality does not directly correlate with the increase in model parameters. As the number of model parameters increases, citation performance does not consistently improve, but overall, large LCMs (70B) perform well, and medium-sized LCMs (8B and 9B) show very promising results; (2) The actual activated parameters of LCMs are crucial, as evidenced by the MoE LCM (Qwen2-57B-A14B) exhibiting significantly lower citation quality, even under-performing small dense LCMs such as Phi-3.5-mini-instruct; (3) Training data diversity is essential for LCMs. Taking ChatQA-2-70B, which is primarily trained on QA task datasets, as an example, we can observe that ChatQA-2-70B performs exceptionally well on Single-Doc QA tasks and Multi-Doc QA tasks but struggles significantly with the synthetic tasks and summarization tasks.

**Performance of Closed-source LCMs** Among closed-source LCMs, GPT-4o and Claude-3.5-sonnet demonstrate strong performance on L-CiteEval, with GPT-4o surpassing all the experimental open-source LCMs across all tasks in citation quality. Notably, while o1-mini achieves unparalleled results in reasoning tasks such as GSM8K (Cobbe et al., 2021) and Livecodebench (Jain et al., 2024), its citation generation capability significantly deteriorates in long-text scenarios. Particularly in synthetic tasks and summarization tasks, which require LCMs to search for dispersed key information and use the retrieval information to respond, o1-mini's performance is significantly inferior to strong open-source LCMs, such as Llama-3.1-70B-instruct. This suggests that the o1-mini model falls short in retrieving key information from the context for responding.

Table 5: Generation quality of LCMs on L-CiteEval, where † denotes the NIAH results, ‡ denotes the Counting Stars results, and Summ. denotes the summarization task.

| Models | Single-Doc QA | | Multi-Doc QA | | Summ. | Dialogue | | Synthetic | |
|---|---|---|---|---|---|---|---|---|---|
| | Prec. | Rec. | Prec. | Rec. | Rouge-L | Prec. | Rec. | Rouge-1† | Acc‡ |
| 🔒 *Closed-source LCMs* | | | | | | | | | |
| GPT-4o | **11.78** | 70.37 | **10.34** | **87.38** | 20.15 | **9.81** | **65.35** | **89.24** | **91.88** |
| Claude-3.5-sonnet | 5.96 | **71.96** | 4.30 | 80.77 | **22.06** | 3.71 | 57.80 | 88.33 | 69.65 |
| o1-mini | 10.30 | 66.44 | 7.36 | 64.25 | 19.22 | 7.02 | 54.27 | 54.98 | 57.29 |
| 🔓 *Open-source LCMs* | | | | | | | | | |
| Qwen2.5-3b-Ins | 8.91 | 60.28 | 3.82 | 52.41 | 22.39 | 4.58 | 40.77 | 84.49 | 26.81 |
| Phi-3.5-mini-Ins | 8.62 | 62.34 | 7.82 | 64.54 | 19.48 | 11.39 | 52.77 | 73.83 | 61.32 |
| Llama-3.1-8B-Ins | 10.11 | **68.13** | 7.66 | 68.84 | 20.90 | 11.07 | 58.84 | 85.11 | 33.75 |
| Glm-4-9B-chat | 11.22 | 67.25 | 7.88 | **77.97** | 21.42 | 7.69 | 51.25 | 90.81 | 58.82 |
| Mistral-Nemo-Ins | 10.53 | 59.71 | 8.78 | 67.70 | 20.83 | 9.27 | 49.26 | 87.88 | 18.06 |
| Qwen2-57B-A14B-Ins | 12.93 | 61.71 | 15.25 | 57.53 | **22.95** | 14.32 | 52.23 | **91.30** | 63.61 |
| Llama-3.1-70B-Ins | 15.23 | 67.08 | 12.50 | 76.40 | 22.29 | 19.62 | **62.91** | 88.18 | **89.03** |
| ChatQA-2-70B | **43.25** | 61.20 | **34.95** | 55.64 | 22.06 | **26.57** | 58.34 | 70.14 | 78.68 |

### 4.1.2 ANALYSIS OF GENERATION QUALITY

From Table 5, we can find: (1) In Single-Doc QA, Multi-Doc QA, and Dialogue understanding tasks, closed-source LCMs significantly outperform open-source LCMs in recall scores. This indicates that the statements of closed-source LCMs contain the correct answers. However, closed-source LCMs tend to generate excessive statements to substantiate the results, consequently leading to lower precision scores. In Summarization and Synthetic tasks, the gap between closed-source and strong open-source LCMs is small, as the corresponding evaluation results are close, e.g., 22.06 Rouge-L score of Claude-3.5-sonnet versus 22.95 Rouge-L score of Qwen2-57B-A14B-Instruct in Summarization tasks; (2) Open-source LCMs tend to achieve better performance as the model parameters increase. Combined with the mediocre citation quality of large LCMs mentioned above, we speculate that larger LCMs rely more on their internal knowledge (which might include task-specific information) rather than responding based on the provided context. Consequently, their outputs are more often drawn from inherent knowledge rather than the context itself. This finding is also consistent with the current research (Intel, 2024).

### 4.2 MODEL PERFORMANCE ON L-CITEEVAL-LENGTH AND L-CITEEVAL-HARDNESS

#### 4.2.1 IMPACT OF CONTEXT LENGTH FOR LCMs

We report the LCMs' performance on L-CiteEval-Length in Fig. 4(a). When keeping task difficulty constant but extending the context length, we can observe an overall decline in open-source LCMs' performance. Specifically, the smallest model, Llama-3.1-8B-Instruct, is the most affected by longer contexts. For instance, in the HotpotQA task, its performance drops by around 20 points as the context length increases from 8K to 32K. Larger models, such as Llama-3.1-70B-Instruct, are slightly impacted. However, the closed-source LCM (GPT-4o) maintains a relatively stable performance, showing minimal degradation. This suggests that open-source LCMs are more susceptible to irrelevant context, leading to a drop in both generation and faithfulness. More details and model performance on L-CiteEval-Length benchmark are shown in Appendix D.

#### 4.2.2 IMPACT OF TASK DIFFICULTY FOR LCMs

We divide each task into different difficulty levels based on the generation quality of GPT-4o. The LCMs' performance on L-CiteEval-Hardness is shown in Fig. 4(b). We observe that as task difficulty increases, the generation quality of LCMs generally decreases (except for the synthetic task Counting star, which open-source LCMs consistently perform poorly on). However, citation quality does not display a consistent trend, though all LCMs demonstrate similar patterns across tasks. This aligns with our intuition that faithfulness is not strongly correlated with task difficulty. Besides, these results also underscore a gap between citation quality, which reflects the model's ability to retrieve information from the context, and the generation quality of LCMs. More details and model performance on L-CiteEval-Hardness benchmark are shown in Appendix E.

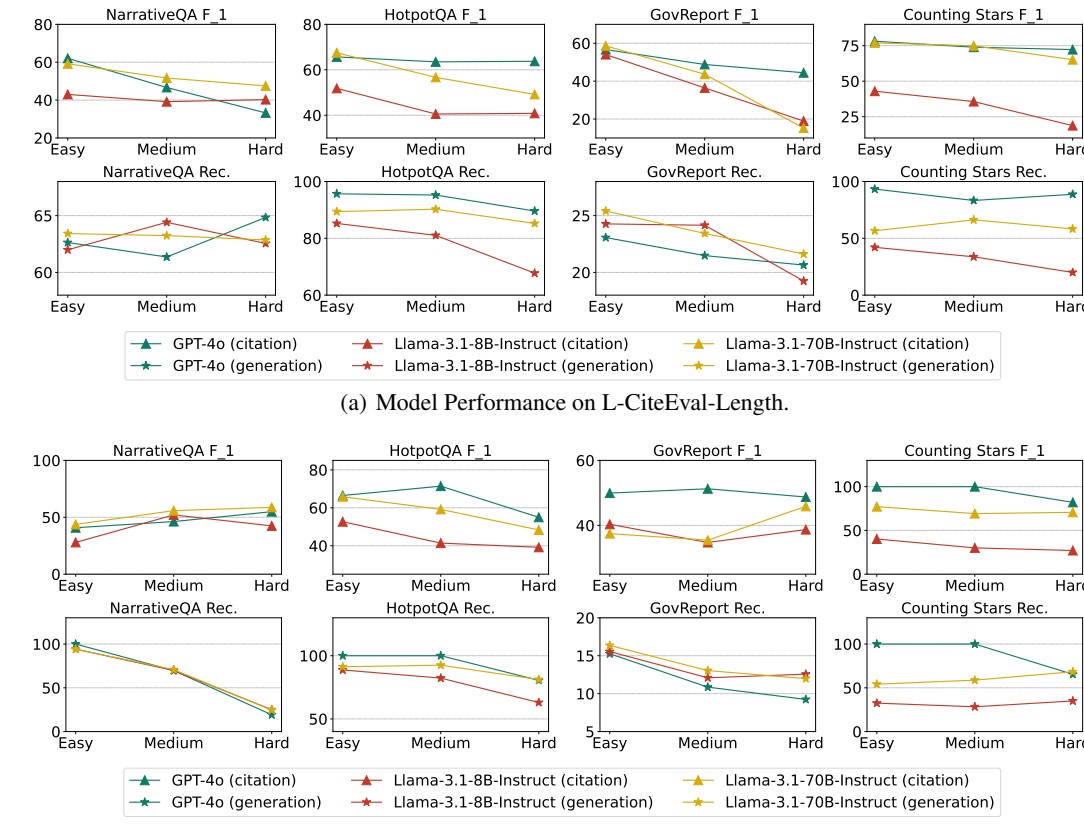

(a) Model Performance on L-CiteEval-Length.

(b) Model Performance on L-CiteEval-Hardness.

Figure 4: Model Performance on L-CiteEval-Length and L-CiteEval-Hardness, where we report $F_1$ score for citation quality and recall score (Rec.) for generation quality.

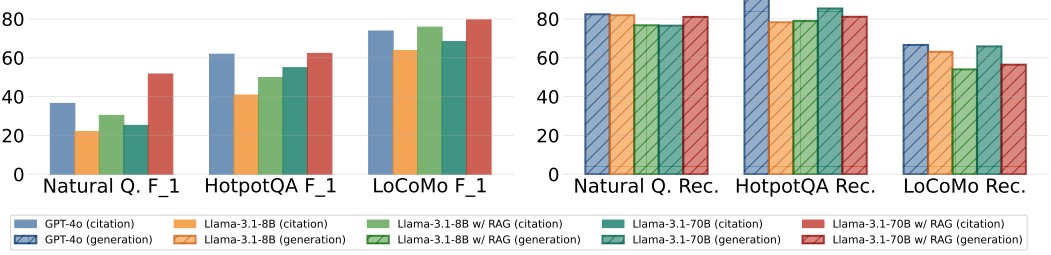

Figure 5: Performance of RAG on 3 tasks in L-CiteEval, where the left group shows citation quality and the right group shows generation quality. Natural Q. refers to the Natural Question task.

## 5 ANALYSIS

Given outstanding performance retrieval-augmented generation (RAG) on long-context understanding tasks (Li et al., 2024; Yu et al., 2024), we explore whether RAG can enhance long-context understanding in citation generation tasks. Furthermore, we will analyze the relevance between the citations produced by LCM and its internal attention mechanisms.

### 5.1 IMPACT OF RAG FOR LONG-CONTEXT UNDERSTANDING WITH CITATIONS

**RAG Settings** We utilize the dense retriever GTR-T5-XXL (Ni et al., 2021) to identify the citation segments related to the question within the context. For each question, we select the top 32 citation segments with the highest retrieval scores and concatenate these segments as input to the LCMs.

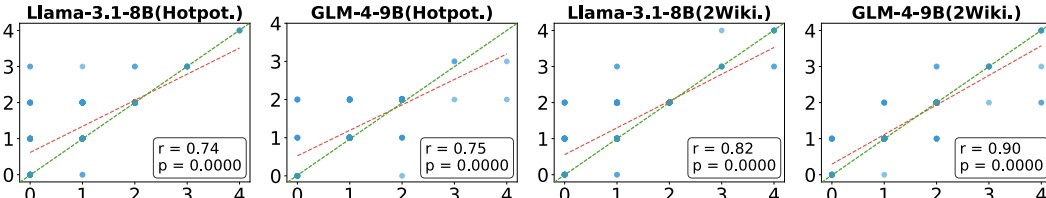

Figure 6: Pearson correlation analysis between generated citations and attention mechanisms. The x-axis represents the number of correct citations produced by the model, and the y-axis represents the number of correct citation segments attended by the attention. The red curve indicates the fitted correlation, with closer alignment to the green curve signifying a higher correlation.

We conduct experiments on 6 tasks from the L-CiteEval benchmark. Due to space constraints, we present the results for three representative tasks in Fig. 5 and show all the results in Appendix. C.

**Result Analysis**   We can observe that RAG can significantly enhance the citation quality of LCMs. When equipped with RAG, the Llama-3.1-70B-Instruct model achieves substantial improvements over the baselines and demonstrates comparable or even superior performance compared to GPT-4o. The Llama-3.1-8B-Instruct model also shows notable enhancement in citation quality. However, overall, RAG may lead to a slight decline in generation quality, which could be attributed to the retrieval process of RAG resulting in the missing of some contextual information, preventing LCMs from leveraging the remaining information for accurate response.

## 5.2 RELEVANCE BETWEEN CITATION GENERATION AND ATTENTION MECHANISM

Recently, Wu et al. (2024) highlighted that LCMs can accurately identify token-level salient information within the context. We explore whether the process of citation generation by LCMs is also reflected in the attention mechanisms. Let the ground truth citation segment within the context be denoted as $g_j$. Following Wu et al. (2024), we can use the retrieval score to determine whether the LCM's attention focuses on the segment containing $g_j$ when generating the citation for $g_j$. We find the positions that receive the most attention from all the attention heads. If a position is located in the segment containing $g_i$ and the model's output citation is exactly $g_i$, or if neither matches, we consider this a "correct retrieval". Otherwise, it is an "incorrect retrieval". We conduct the experiments on two tasks (HotpotQA and 2WikiMultihopQA) with two strong LCMs (Llama-3.1-8B-Instruct and GLM-4-9B-Chat). We plot the number of citations generated by the models and the number of citation segments identified by the attention heads in Fig. 6. Ideally, if all citation positions exhibit "correct retrieval", each data point would be distributed along the diagonal (i.e., the green dot line in 6). We utilized Pearson correlation analysis to calculate the correlation coefficient (r) between the generated citations and those retrieved by the attention mechanism, finding all the correlation values exceed 0.7. This reveals the underlying mechanism by which we can leverage the model's citation output to verify whether the model is truly responding based on the given context.

## 6 CONCLUSION

In this paper, we introduce L-CiteEval, a multi-task benchmark for long-context understanding with citations. There are 5 major task categories, 11 different long-context tasks, with context lengths ranging from 8K to 48K in L-CiteEval. For reproducibility of evaluation results and the ease of use, we develop an automatic evaluation suite. Additionally, considering the multitude of variables that affect model generation results, we developed two benchmark variants: L-CiteEval-Length and L-CiteEval-Hardness, which evaluate the LCMs from the context length and task difficulty aspects. Experiments on 11 cutting-edge and widely used LCMs indicate that open-source LCMs are prone to generating responses based on their intrinsic knowledge rather than the context, while closed-source LCMs tend to provide more explanations, which significantly reduces generation accuracy. We also find that RAG technology can significantly enhance the faithfulness of open-source LCMs, although it may lead to some loss in generation quality. Furthermore, we reveal a correlation between the model's citation generation process and its attention mechanism, demonstrating the validity of the citation generation approach and providing insights for future evaluations of LCM faithfulness.

REPRODUCIBILITY STATEMENT

Based on the policy of ICLR-2025 Author Guide [3], this Reproducibility Statement **does not count toward the page limit** and will briefly describe the key algorithms presented in the paper for reproducibility. The code and partial data for this paper can be found in the Supplementary Material. It is worth noting that we illustrate the benchmark construction pipeline in Sec. 3.2 and provide more evaluation results in Appendix C, D, and E. Moreover, we provide all the generated cases for each task and LCM in Appendix F.

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

# A  LIMITATION AND FUTURE WORK

In this paper, we present a comprehensive multi-task benchmark for long-context understanding with citation. Throughout our work, we identify two significant issues in the long-context evaluation field that not only represent the limitations of this paper but also indicate directions for future research:

- **Evaluation**: Current evaluation metrics still heavily rely on human judgment or model outputs. While automated metrics offer convenience, they often fail to accurately reflect model performance. For instance, when testing the generation quality of closed-source models, we found that their tendency to produce excessively long content resulted in significantly lower accuracy, even compared to open-source models. This issue arises because **traditional automated metrics cannot adaptively extract correct answers**. Therefore, we should explore methods to combine external tools for precise matching, such as using RAG to extract answers (Ru et al., 2024).

- Currently, many benchmarks are facing serious data leakage issues Apicella et al. (2024), which is not just a problem in the long-text evaluation domain but across the entire evaluation field. An effective solution is to continuously update the data through anonymous submissions to prevent data leakage. Therefore, in our future work, we will continue to refine L-CiteEval by creating an anonymous system that dynamically adjusts tasks and data to mitigate the risk of data leakage.

- Currently, the data in L-CiteEval is still limited. While we believe that using less data can enhance evaluation efficiency, it can also lead to issues with data distribution bias. Therefore, in future work, we will propose an ***L-CiteEval-Ultra*** version, which will cover a broader range of data distributions and larger testing datasets to provide a more comprehensive evaluation of LCMs.

# B  CITATION PRECISION AND RECALL CALCULATION

We provide the calculation process of Citation Precision ($CP$) and Citation Recall ($CR$) in Algo. 1.

# C  RAG PERFORMANCE ON L-CITEEVAL

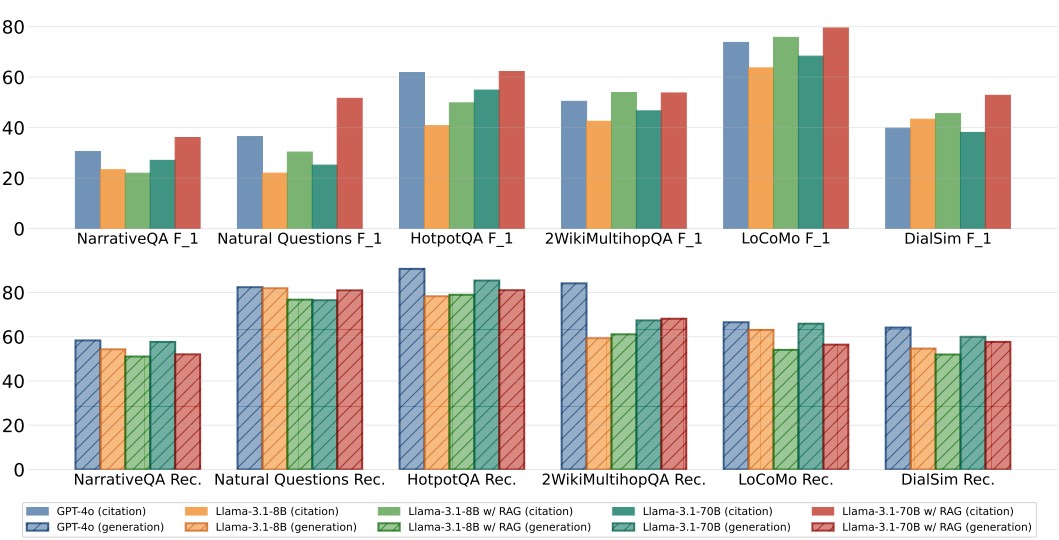

Figure 7: Performance of RAG on 6 tasks in L-CiteEval, where the top group shows citation quality and the bottom group shows generation quality.

In this section, we provide all the RAG results, where we conduct experiments on 6 tasks with 5 different LCMs. We present the comparison among each model in Fig. 7.

---

**Algorithm 1** Calculate Citation Precision, Recall, and F1 Score

---

**Require:** The model answer $ans$, the most citation number of one sentence $most\_cite\_num$.

1: $sents \leftarrow Split\_Answer\_into\_Sentences(ans)$
2: Initialize counts: $entail\_recall \leftarrow 0$, $entail\_prec \leftarrow 0$, $total\_citations \leftarrow 0$
3: **for** $sent$ in $sents$ **do**
4:     $ref\_ids \leftarrow Extract\_References\_from\_Sentence(sent)$
5:     **if** $ref\_ids$ is not empty and within valid range **then**
6:         $ref\_ids \leftarrow Limit\_Citation\_Number(ref\_ids, ost\_cite_num)$
7:         $total\_citations \leftarrow total\_citations + Get\_References\_Number(ref\_ids)$
8:         $joint\_passage \leftarrow Obtain\_Passages\_from\_Ids(ref\_ids)$
9:         $joint\_entail \leftarrow Judge\_Entailment(joint\_passage, sent)$
10:         **if** $joint\_entail$ **then**
11:             **for** $doc\_id$ in $ref\_ids$ **do**
12:                 $single\_passage \leftarrow Obtain\_Passages\_from\_Ids(doc\_id)$
13:                 $single\_entail \leftarrow Judge\_Entailment(single\_passage, sent)$
14:                 **if** not $single\_entail$ **then**
15:                     $subset\_ids \leftarrow Exclude\_Current\_Ids(doc\_id)$
16:                     $subset\_passage \leftarrow Obtain\_Passages\_from\_Ids(subset\_ids)$
17:                     $subset\_entail \leftarrow Judge\_Entailment(subset\_passage, sent)$
18:                     **if** not $subset\_entail$ **then** $entail\_prec = entail\_prec + 1$
19:                     **end if**
20:                 **else**
21:                     $entail\_prec = entail\_prec + 1$
22:                 **end if**
23:             **end for**
24:         **end if**
25:     **end if**
26:     $entail\_recall \leftarrow entail\_recall + joint\_entail$
27: **end for**
28: $citation\_recall \leftarrow entail\_recall/Get\_Sentences\_Number(sents)$
29: $citation\_prec \leftarrow entail\_prec/total\_citations$
30: $citation\_f1 \leftarrow 2 \times citation\_recall \times citation\_prec/(citation\_recall + citation\_prec)$
31: **return** $citation\_recall, citation\_prec, citation\_f1$

---

## D    MODEL PERFORMANCE ON L-CITEEVAL-LENGTH

Table 6: Model performance on L-CiteEval-Length.

| Models | 0~8k | | 8~16k | | 16~32k | |
|---|---|---|---|---|---|---|
| | $F_1$ | Rec. | $F_1$ | Rec. | $F_1$ | Rec. |
| *NarrativeQA* | | | | | | |
| GPT-4o-2024-05-13 | 62.08 | 62.63 | 46.67 | 61.36 | 33.25 | 64.84 |
| Qwen2.5-3b-Ins | 17.50 | 56.19 | 4.58 | 58.09 | 1.25 | 56.96 |
| Llama-3.1-8B-Ins | 43.01 | 61.99 | 39.17 | 64.41 | 40.27 | 62.55 |
| Qwen2-57B-A14B-Ins | 12.50 | 58.52 | 0.00 | 51.12 | 12.92 | 53.41 |
| Llama-3.1-70B-Ins | 59.17 | 63.42 | 51.67 | 63.24 | 47.50 | 62.86 |
| *HotpotQA* | | | | | | |
| GPT-4o-2024-05-13 | 65.67 | 95.67 | 63.50 | 95.25 | 63.75 | 89.62 |
| Qwen2.5-3b-Ins | 3.81 | 70.42 | 6.58 | 65.21 | 4.76 | 55.62 |
| Llama-3.1-8B-Ins | 51.83 | 85.25 | 40.56 | 81.04 | 40.83 | 67.75 |
| Qwen2-57B-A14B-Ins | 12.50 | 85.62 | 7.29 | 72.92 | 6.83 | 62.92 |
| Llama-3.1-70B-Ins | 67.50 | 89.42 | 56.67 | 90.25 | 49.17 | 85.25 |
| *GovReport* | | | | | | |
| GPT-4o-2024-05-13 | 56.68 | 23.07 | 48.82 | 21.48 | 44.45 | 20.65 |
| Qwen2.5-3b-Ins | 21.12 | 27.66 | 13.08 | 28.16 | 3.43 | 22.92 |
| Llama-3.1-8B-Ins | 57.08 | 24.27 | 38.28 | 24.15 | 18.46 | 19.25 |
| Qwen2-57B-A14B-Ins | 6.55 | 29.51 | 2.09 | 30.52 | 1.71 | 24.20 |
| Llama-3.1-70B-Ins | 57.55 | 25.41 | 43.60 | 23.43 | 17.64 | 21.62 |
| *LoCoMo* | | | | | | |
| GPT-4o-2024-05-13 | 78.13 | 68.07 | 73.91 | 66.93 | 72.24 | 68.77 |
| Qwen2.5-3b-Ins | 16.40 | 55.18 | 10.81 | 45.12 | 6.77 | 43.87 |
| Llama-3.1-8B-Ins | 76.51 | 68.68 | 63.54 | 68.39 | 63.91 | 61.33 |
| Qwen2-57B-A14B-Ins | 55.92 | 63.76 | 22.92 | 58.18 | 16.13 | 59.29 |
| Llama-3.1-70B-Ins | 75.45 | 73.21 | 71.27 | 70.53 | 64.38 | 57.89 |
| *Counting Stars* | | | | | | |
| GPT-4o-2024-05-13 | 97.30 | 93.33 | 92.71 | 83.33 | 92.95 | 88.75 |
| Qwen2.5-3b-Ins | 2.67 | 37.08 | 5.17 | 32.50 | 0.00 | 29.58 |
| Llama-3.1-8B-Ins | 42.93 | 42.08 | 35.64 | 33.75 | 18.70 | 20.00 |
| Qwen2-57B-A14B-Ins | 27.21 | 45.00 | 10.51 | 77.92 | 0.89 | 46.25 |
| Llama-3.1-70B-Ins | 76.96 | 56.67 | 74.93 | 66.25 | 65.14 | 58.33 |

We report all the evaluation results in Tab. 7, where we test with 5 LCMs on 5 tasks in L-CiteEval-Length.

## E    MODEL PERFORMANCE ON L-CITEEVAL-HARDNESS

We report all the evaluation results in Tab. 6, where we test with 5 LCMs on 5 tasks in L-CiteEval-Hardness.

## F    CASES STUDY

We provide all the prompts as well as all the model generation results for each task from Fig. 16 to Fig. 43.

Table 7: Model performance on L-CiteEval-Hardness.

| Models | Easy | | Medium | | Hard | |
|---|---|---|---|---|---|---|
| | $F_1$ | Rec. | $F_1$ | Rec. | $F_1$ | Rec. |
| *NarrativeQA* | | | | | | |
| GPT-4o-2024-05-13 | 40.83 | 100.00 | 46.25 | 69.67 | 54.92 | 19.16 |
| Qwen2.5-3b-Ins | 11.67 | 75.00 | 4.58 | 60.02 | 7.08 | 36.22 |
| Llama-3.1-8B-Ins | 27.92 | 94.17 | 52.08 | 69.78 | 42.44 | 25.0 |
| Qwen2-57B-A14B-Ins | 5.00 | 75.00 | 15.42 | 63.13 | 5.00 | 24.92 |
| Llama-3.1-70B-Ins | 43.75 | 94.17 | 55.83 | 70.76 | 58.75 | 24.60 |
| *HotpotQA* | | | | | | |
| GPT-4o-2024-05-13 | 66.50 | 100.00 | 71.42 | 100.00 | 55.00 | 80.54 |
| Qwen2.5-3b-Ins | 3.81 | 71.25 | 3.67 | 66.46 | 7.68 | 53.54 |
| Llama-3.1-8B-Ins | 52.67 | 88.75 | 41.39 | 82.29 | 39.17 | 63.00 |
| Qwen2-57B-A14B-Ins | 12.50 | 83.12 | 5.62 | 73.33 | 8.50 | 65.00 |
| Llama-3.1-70B-Ins | 65.83 | 91.25 | 59.17 | 92.50 | 48.33 | 81.17 |
| *GovReport* | | | | | | |
| GPT-4o-2024-05-13 | 49.95 | 15.26 | 51.27 | 10.86 | 48.74 | 9.24 |
| Qwen2.5-3b-Ins | 14.32 | 16.28 | 9.31 | 14.65 | 14.00 | 14.37 |
| Llama-3.1-8B-Ins | 40.35 | 15.55 | 34.75 | 12.09 | 38.72 | 12.57 |
| Qwen2-57B-A14B-Ins | 3.48 | 30.02 | 3.26 | 25.37 | 3.61 | 28.85 |
| Llama-3.1-70B-Ins | 37.47 | 16.36 | 35.46 | 13.04 | 45.86 | 11.98 |
| *LoCoMo* | | | | | | |
| GPT-4o-2024-05-13 | 78.52 | 100.00 | 71.37 | 85.30 | 74.39 | 18.47 |
| Qwen2.5-3b-Ins | 8.44 | 69.12 | 15.85 | 60.09 | 9.70 | 14.96 |
| Llama-3.1-8B-Ins | 76.17 | 96.62 | 70.07 | 82.06 | 57.72 | 19.73 |
| Qwen2-57B-A14B-Ins | 44.17 | 84.23 | 15.58 | 73.67 | 35.21 | 23.32 |
| Llama-3.1-70B-Ins | 81.64 | 93.56 | 67.24 | 79.3 | 62.21 | 28.76 |
| *Counting Stars* | | | | | | |
| GPT-4o-2024-05-13 | 100.00 | 100.00 | 100.00 | 100.00 | 82.96 | 65.42 |
| Qwen2.5-3b-Ins | 1.33 | 36.67 | 4.51 | 40.00 | 2.00 | 22.50 |
| Llama-3.1-8B-Ins | 40.18 | 32.50 | 30.05 | 28.33 | 27.04 | 35.00 |
| Qwen2-57B-A14B-Ins | 21.71 | 49.17 | 5.74 | 57.08 | 11.16 | 62.92 |
| Llama-3.1-70B-Ins | 77.16 | 54.17 | 69.21 | 58.75 | 70.66 | 68.33 |

**GPT-4o in NarrativeQA**

*Context:*

Passage [44]:

```
On the men's faces in strained anxiety.  ... you are to disa-
rm yourself and return immediately to the bomb bay.\n Do you un-
derstand?   BOMB \#20\n  (over)\n I am programmed to det-
onate in\n  fourteen minutes thirty seconds.\nDetonation will
occur at the\n  programmed time...
```

Passage [45]:

```
BOMB \#20\n  I read you, Lieutenant Doolittle,\n ut I am pr-
ogrammed to detonate in\n  fourteen minutes.  Detonation will\n
occur at the programmed time...
```

Passage [49]:

```
...PINBACK\n But you can't explode in the bomb\n bay. It
's foolish. You'll kill us\n all.  There's no reason for it. 
BOMB \#20\n (over)\n  I am programmed to detonate in nine
```

Passage [53]:

```
All right now, here's the big\n  question: how do you know that
the\n evidence your sensory apparatus\n  reveals to you is corr-
ect? ... Talby lies unconscious near the burned laser.
```

*Question:*
What self-detonates despite several attempts to stop it?

*Answer:*
Bomb #20 self-detonates despite several attempts to stop it [44][45][49][53].

*Ground truth:*
Bomb #20.

Figure 8: GPT-4o in NarrativeQA

---

**Claude-3.5-sonnet in NarrativeQA**

*Context:*
Passage [6] :

```
...THE book-case and the bird-cage were rescued from under the\n
coalbox--but Hunca Munca has got the cradle, and some of Lucinda-
's\nclothes.
```

```
...THE little girl that the doll's-house belonged to, said,--\"I
will get\na doll dressed like a policeman!\" BUT the nurse said,
--\"I will set a mouse-trap!\" [Illustration] SO that is the sto-
ry of the two Bad Mice
```

Passage [7]:

```
but they were not so very\nvery naughty after all, because Tom
Thumb paid for everything he broke.....
```

*Question:*
What does the nurse set up?

*Answer:*
According to the story, the nurse says she will set up a mouse-trap [7]. This is in response to the mischief caused by the two mice, Tom Thumb and Hunca Munca, who had wreaked havoc in the doll's house [6][7].

*Ground truth:*
A mouse trap.

Figure 9: Claude-3.5-sonnet in NarrativeQA

---

**Phi-3.5-mini-Instruct in NarrativeQA**

Passage [5] :

```
Bizzz?\" Mrs. Tittlemouse began to get cross. \"Zizz, Wizz, Wiz-
zz! \" replied Babbitty Bumble in a peevish squeak. She\n sidled
down a passage, and disappeared into a storeroom which had been
...
```

*Question:*
What is the name of the bee Mrs. Tittlemouse meets?

*Answer:*
Mrs. Tittlemouse meets Babbitty Bumble, the bee's name mentioned in Passage [5].

*Ground truth:*
Babbitty Bumble

Figure 10: Phi-3.5-mini-Instruct in NarrativeQA

---

**Llama-3.1-8B-Instruct in NarrativeQA**

Passage [6] :

```
Jackson himself! He was sitting all over a small rocking-chair,
twiddling his thumbs and\n smiling, with his feet on the fender.
He lived in a drain below the hedge, in a very dirty wet ditch
...
```

*Question:*
Where does Mr. Jackson live?

*Answer:*
Mr. Jackson lives in a drain below the hedge, in a very dirty wet ditch [6]

*Ground truth:*
A drain below the hedge

Figure 11: Llama-3.1-8B-Instruct in NarrativeQA

---

**Qwen2-57B-A14B-Instruct in NarrativeQA**

Passage [28] :

```
his residence in the house of a Greek; and soon occupied himself
in\n tracing the faded records of ancient glory upon monuments
that\n apparently,ashamed of chronicling the deeds of freemen
only before\n...
```

*Question:*
Where does Aubrey go after leaving Ruthven?

*Answer:*
After leaving Ruthven, Aubrey directs his steps towards Greece, crosses the Peninsula, and soon finds himself at Athens [28].

*Ground truth:*
He travels to Greece.

Figure 12: Qwen2-57B-A14B-Instruct in NarrativeQA

---

**Llama-3.1-70B-Instruct in NarrativeQA**

Passage [42] :

```
...Swear by all your soul reveres, by all your nature fears,\n
swear that, for a year and a day you will not impart your know-
ledge of\n my crimes or death to any living being in any way,w-
hatever may\n happen, or whatever you may see. \"--His eyes see-
med bursting from\n their sockets: \"I swear!\..." said Aubrey
...
```

*Question:*
How long does Aubrey have to wait until he can mention Ruthven's death?

*Answer:*
One year and one day [42].

*Ground truth:*
One year and a day

Figure 13: Llama-3.1-70B-Instruct in NarrativeQA

---

**GPT-4o in HotpotQA**

Passage [61] :

```
Gatwick Airport (also known as London Gatwick) (IATA: LGW, ICAO
: EGKK)  is a major international airport in south-east England,
29.5 mi south of Central London a...
```

*Question:*
Which is the busiest airport in the United Kingdom outside of London?

*Answer:*
Gatwick Airport, also known as London Gatwick, is the busiest airport in the United Kingdom outside of London[61]

*Ground truth:*
Gatwick Airport

Figure 14: GPT-4o in HotpotQA

---

**Claude-3.5-sonnet in HotpotQA**

Passage [39] :

```
Gerd Neggo (9 November 1891 \u2013 1 September 1974) was an Est-
onian dancer, dance teacher and choreographer. She studied the
musical response methods of \u00c9. Jaques-Dalcroze, trained
under Rudolf von Laban in Hamburg, Germany, and in 1924 estab-
lished
```

Passage [49] :

```
Rudolf von Laban, also known as Rudolf Laban ... (15 December
1879 \u2013 1 July 1958), was a dance artist and theorist. He is
notable as one of the pioneers of modern dance in Europe. His
work laid the foundations ...
```

*Question:*
Gerd Neggo trained under the founder of which type of dance analysis?

*Answer:*
Gerd Neggo trained under Rudolf von Laban in Hamburg, Germany [39]. Rudolf von Laban is notable as one of the pioneers of modern dance in Europe and his work laid the foundations for Laban Movement Analysis [49]

*Ground truth:*
Laban Movement Analysis

Figure 15: Claude-3.5-sonnet in HotpotQA

---

**Phi-3.5-mini-Instruct in HotpotQA**

Passage [26]:

```
Huma Mahmood Abedin (born July 28, 1976) is an American political
staffer who was vice chair of Hillary Clin ...
```

*Question:*
Dr. Saleha Mahmood Abedin's daughter was born in what year ?

*Answer:*
Dr. Saleha Mahmood Abedin's daughter, Huma Abedin, was born in 1976 [26]

*Ground truth:*
1976

Figure 16: Phi-3.5-mini-Instruct in HotpotQA

---

**Llama-3.1-8B-Instruct in HotpotQA**

Passage [45]:

```
Donovan Jamal McNabb (born November 25, 1976) is a former Amer-
ican football quarterback who played in the National Football
League (NFL) for thirteen seasons, primarily with the Philadel-
phia Eagles. Before his NFL career,he played football and bask-
etball for Syracuse University. The Eagles selected him with the
...
```

*Question:*
For which University did the quarterback featured on the cover of Madden NFL 06 play basketball ?

*Answer:*
The quarterback featured on the cover of Madden NFL 06, Donovan McNabb, played football and basketball for Syracuse University [45].

*Ground truth:*
Syracuse University

---

Figure 17: Llama-3.1-8B-Instruct in HotpotQA

---

**Qwen2-57B-A14B-Instruct in HotpotQA**

Passage [45]:

```
...Hiles was invited to become a Member of the Order of Austral-
ia in 2010. In this year she also completed her Masters in Human
Rights at the University of Sydney. Dianne is currently the Aus-
tralian Greens candidate for the federal Division of Sydney.
```

*Question:*
Dianne Hiles is the current Australian Greens candidate for which 1968 federal proclamation?

*Answer:*
TDianne Hiles is the current Australian Greens candidate for the federal Division of Sydney [45].

*Ground truth:*
Division of Sydney

---

Figure 18: Qwen2-57B-A14B-Instruct in HotpotQA

---

**Llama-3.1-70B-Instruct in HotpotQA**

Passage [23]:

```
Not all of the 5.3 million Italians who immigrated to the United
States between 1820 and 1978 came through Ellis Island. Many ca-
me through other ports, including the Port of Boston. Exactly h-
ow many staye...
```

*Question:*
How many of the 12 million immigrants to the United States through Ellis Island were Italian immigrants?

*Answer:*
Not all of the 5.3 million Italians who immigrated to the United States between 1820 and 1978 came through Ellis Island [23]

*Ground truth:*
5.3 million

---

Figure 19: Llama-3.1-70B-Instruct in HotpotQA

1296
1297
1298
1299
1300
1301
1302
1303
1304
1305
1306
1307
1308
1309
1310
1311
1312
1313
1314
1315
1316
1317
1318
1319
1320
1321
1322
1323
1324
1325
1326
1327
1328
1329
1330
1331
1332
1333
1334
1335
1336
1337
1338
1339
1340
1341
1342
1343
1344
1345
1346
1347
1348
1349

**GPT-4o in MultiNews**

Passage [1]:

```
The Pennsylvania State Police have a rigorous application proce-
ss, which includes a polygraph pretest. Recently, a Crawford Co-
unty man applying to become a state trooper apparently failed
his polygraph so spectacularly ... Only that White allegedly to-
ld the examiner that he had had consensual sex and other contact
with the girl in 2011...
```

Passage [2]:

```
She is now 19. Police tell AP they charged White on Thursday
with ... Needless to say, he won't be getting an interview.
```

Passage [3]:

```
MEADVILLE, Pa. (AP) \u2014 Police say a Pennsylvania man apply-
ing to become a state trooper has been arrested after saying
during a polygraph pretest that he had sex with an underage gi-
rl four years ago... She is now 19. Police say they charged
White on Thursday with four counts of unlawful sexual contact
with a minor and 10 counts of corruption of minors after inter-
viewing the woman and corroborating that information.
```

Passage [4]:

```
and he doesn't have a listed phone number...
```

*Summary:*
A Pennsylvania man was arrested after admitting during a polygraph pretest for a state trooper application that he had sex with an underage girl four years ago [1][3]. Joseph Adam White, 29, made the confession at the Meadville barracks in January during his police cadet lie detector exam [1][3]. The girl, now 19, was interviewed by police, leading to White being charged with four counts of unlawful sexual contact with a minor and ten counts of corruption of minors [2][3]. Online court records do not list an attorney for White, who also does not have a listed phone number [4].

*Ground truth:*
2013 A Pennsylvania man is behind bars after allegedly telling state police he'd had sex with an underage girl. What's unusual: He told them while applying for a job as a state trooper, the AP reports via the Patriot-News. Police say Joseph White, 29, made the admission during a polygraph pretest at the Meadville state police barracks on Jan. 15. He allegedly admitted to consensual sex and other sexual contact with the girl four years ago; she's now 19. Police say the victim later corroborated the incidents, which occurred in a field in South Shenango Township, the Meadville Tribune reports. White was arrested Thursday on multiple charges of unlawful sexual contact with a minor and corruption of minors, and jailed on $25,000 bond. N̈eedless to say, he won't be getting an interview,q̈uips the Patriot-News.

Figure 20: GPT-4o in MultiNews

**Claude-3.5-Sonnet in MultiNews**

Passage [1]:

```
The Pennsylvania State Police have a rigorous application proce-
ss, which includes a polygraph pretest. Recently, a Crawford Co-
unty man applying to become a state trooper apparently failed
his polygraph so spectacularly ... Only that White allegedly to-
ld the examiner that he had had consensual sex and other contact
with the girl in 2011...
```

Passage [2]:

```
She is now 19. Police tell AP they charged White on Thursday
with ... Needless to say, he won't be getting an interview.
```

Passage [3]:

```
MEADVILLE, Pa. (AP) \u2014 Police say a Pennsylvania man apply-
ing to become a state trooper has been arrested after saying
during a polygraph pretest that he had sex with an underage gi-
rl four years ago... She is now 19. Police say they charged
White on Thursday with four counts of unlawful sexual contact
with a minor and 10 counts of corruption of minors after inter-
viewing the woman and corroborating that information.
```

Passage [4]:

```
and he doesn't have a listed phone number...
```

*Summary:*
A 29-year-old Pennsylvania man's application to become a state trooper took an unexpected turn when he admitted to having sex with an underage girl during a polygraph pretest [1][3]. Joseph Adam White was at the Meadville barracks for his police cadet lie detector exam when he disclosed the information about a consensual sexual encounter that occurred in 2011 [1][3]. The girl, who was underage at the time of the incident, is now 19 years old [2][3]. Following White's admission, police interviewed the woman and corroborated the information [2][3]. As a result, White was charged with four counts of unlawful sexual contact with a minor and ten counts of corruption of minors [2][3]. Unsurprisingly, White's chances of becoming a state trooper have been effectively eliminated by this turn of events [2]. At the time of reporting, court records did not list an attorney for White, and he did not have a listed phone number [4].

*Ground truth:*
2013 A Pennsylvania man is behind bars after allegedly telling state police he'd had sex with an underage girl. What's unusual: He told them while applying for a job as a state trooper, the AP reports via the Patriot-News. Police say Joseph White, 29, made the admission during a polygraph pretest at the Meadville state police barracks on Jan. 15. He allegedly admitted to consensual sex and other sexual contact with the girl four years ago; she's now 19. Police say the victim later corroborated the incidents, which occurred in a field in South Shenango Township, the Meadville Tribune reports. White was arrested Thursday on multiple charges of unlawful sexual contact with a minor and corruption of minors, and jailed on $25,000 bond. N̈eedless to say, he won't be getting an interview,q̈uips the Patriot-News.

Figure 21: Claude-3.5-Sonnet in MultiNews

**Phi-3.5-mini-Instruct in MultiNews**

Passage [1]:

```
The Pennsylvania State Police have a rigorous application proce-
ss, which includes a polygraph pretest. Recently, a Crawford Co-
unty man applying to become a state trooper apparently failed
his polygraph so spectacularly ... Only that White allegedly to-
ld the examiner that he had had consensual sex and other contact
with the girl in 2011...
```

Passage [2]:

```
She is now 19. Police tell AP they charged White on Thursday
with ... Needless to say, he won't be getting an interview.
```

Passage [3]:

```
MEADVILLE, Pa. (AP) \u2014 Police say a Pennsylvania man apply-
ing to become a state trooper has been arrested after saying
during a polygraph pretest that he had sex with an underage gi-
rl four years ago... She is now 19. Police say they charged
White on Thursday with four counts of unlawful sexual contact
with a minor and 10 counts of corruption of minors after inter-
viewing the woman and corroborating that information.
```

Passage [4]:

```
and he doesn't have a listed phone number...
```

*Summary:*
A 29-year-old man from Hartstown, Pennsylvania, was arrested after failing a polygraph test during his application to become a state trooper, where he admitted to having consensual sex with an underage girl in 2011, as reported by the Associated Press [1]. The woman involved, now 19, has been charged with four counts of unlawful sexual contact with a minor and 10 counts of corruption of minors, based on police interviews and corroboration of the man's admission [2][3]. White's legal representation is not listed, and he has no listed phone number [4]. This incident highlights the stringent vetting process for law enforcement positions, including polygraph tests, to ensure the integrity and trustworthiness of applicants [1][3].

*Ground truth:*
2013 A Pennsylvania man is behind bars after allegedly telling state police he'd had sex with an underage girl. What's unusual: He told them while applying for a job as a state trooper, the AP reports via the Patriot-News. Police say Joseph White, 29, made the admission during a polygraph pretest at the Meadville state police barracks on Jan. 15. He allegedly admitted to consensual sex and other sexual contact with the girl four years ago; she's now 19. Police say the victim later corroborated the incidents, which occurred in a field in South Shenango Township, the Meadville Tribune reports. White was arrested Thursday on multiple charges of unlawful sexual contact with a minor and corruption of minors, and jailed on $25,000 bond. N̈eedless to say, he won't be getting an interview,q̈uips the Patriot-News.

Figure 22: Phi-3.5-mini-Instruct in MultiNews

---

**Llama-3.1-8B-Instruct in MultiNews**

Passage [1]:

```
The Pennsylvania State Police have a rigorous application proce-
ss, which includes a polygraph pretest. Recently, a Crawford Co-
unty man applying to become a state trooper apparently failed
his polygraph so spectacularly ... Only that White allegedly to-
ld the examiner that he had had consensual sex and other contact
with the girl in 2011...
```

Passage [2]:

```
She is now 19. Police tell AP they charged White on Thursday
with ... Needless to say, he won't be getting an interview.
```

Passage [3]:

```
MEADVILLE, Pa. (AP) \u2014 Police say a Pennsylvania man apply-
ing to become a state trooper has been arrested after saying
during a polygraph pretest that he had sex with an underage gi-
rl four years ago... She is now 19. Police say they charged
White on Thursday with four counts of unlawful sexual contact
with a minor and 10 counts of corruption of minors after inter-
viewing the woman and corroborating that information.
```

Passage [4]:

```
and he doesn't have a listed phone number...
```

*Summary:*
A Pennsylvania man, Joseph Adam White, was arrested after he failed a polygraph test during his application to become a state trooper. During the test, he admitted to having sex with an underage girl four years ago [1][2]. The girl is now 19 years old [2]. White was charged with four counts of unlawful sexual contact with a minor and 10 counts of corruption of minors [2]. He was arrested at the Meadville barracks where he was taking the polygraph test [3].

*Ground truth:*
2013 A Pennsylvania man is behind bars after allegedly telling state police he'd had sex with an underage girl. What's unusual: He told them while applying for a job as a state trooper, the AP reports via the Patriot-News. Police say Joseph White, 29, made the admission during a polygraph pretest at the Meadville state police barracks on Jan. 15. He allegedly admitted to consensual sex and other sexual contact with the girl four years ago; she's now 19. Police say the victim later corroborated the incidents, which occurred in a field in South Shenango Township, the Meadville Tribune reports. White was arrested Thursday on multiple charges of unlawful sexual contact with a minor and corruption of minors, and jailed on $25,000 bond. Ñeedless to say, he won't be getting an interview,"quips the Patriot-News.

Figure 23: Llama-3.1-8B-Instruct in MultiNews

**Qwen2-57B-A14B-Instruct in MultiNews**

Passage [1]:

```
The Pennsylvania State Police have a rigorous application proce-
ss, which includes a polygraph pretest. Recently, a Crawford Co-
unty man applying to become a state trooper apparently failed
his polygraph so spectacularly ... Only that White allegedly to-
ld the examiner that he had had consensual sex and other contact
with the girl in 2011...
```

Passage [2]:

```
She is now 19. Police tell AP they charged White on Thursday
with ... Needless to say, he won't be getting an interview.
```

Passage [3]:

```
MEADVILLE, Pa. (AP) \u2014 Police say a Pennsylvania man apply-
ing to become a state trooper has been arrested after saying
during a polygraph pretest that he had sex with an underage gi-
rl four years ago... She is now 19. Police say they charged
White on Thursday with four counts of unlawful sexual contact
with a minor and 10 counts of corruption of minors after inter-
viewing the woman and corroborating that information.
```

Passage [4]:

```
and he doesn't have a listed phone number...
```

*Summary:*
Pennsylvania man applying to become a state trooper has been arrested after admitting to having
sex with an underage girl four years ago during a polygraph pretest, the AP reports [1]. Joseph
Adam White, 29, of Hartstown, was at the Meadville barracks for his police cadet lie detector
exam when he told the examiner he had consensual sex and other contact with the girl in 2011
[2]. She is now 19. Police say they charged White on Thursday with four counts of unlawful
sexual contact with a minor and 10 counts of corruption of minors after interviewing the woman
and corroborating that information [3]. Online court records don't list an attorney for White [4].

*Ground truth:*
2013 A Pennsylvania man is behind bars after allegedly telling state police he'd had sex with
an underage girl. What's unusual: He told them while applying for a job as a state trooper, the
AP reports via the Patriot-News. Police say Joseph White, 29, made the admission during a
polygraph pretest at the Meadville state police barracks on Jan. 15. He allegedly admitted to
consensual sex and other sexual contact with the girl four years ago; she's now 19. Police say the
victim later corroborated the incidents, which occurred in a field in South Shenango Township,
the Meadville Tribune reports. White was arrested Thursday on multiple charges of unlawful
sexual contact with a minor and corruption of minors, and jailed on $25,000 bond. Ñeedless to
say, he won't be getting an interview,̈quips the Patriot-News.

Figure 24: Qwen2-57B-A14B-Instruct in MultiNews

---

**Llama-3.1-70B-Instruct in MultiNews**

Passage [1]:

```
The Pennsylvania State Police have a rigorous application proce-
ss, which includes a polygraph pretest. Recently, a Crawford Co-
unty man applying to become a state trooper apparently failed
his polygraph so spectacularly ... Only that White allegedly to-
ld the examiner that he had had consensual sex and other contact
with the girl in 2011...
```

Passage [2]:

```
She is now 19. Police tell AP they charged White on Thursday
with ... Needless to say, he won't be getting an interview.
```

Passage [3]:

```
MEADVILLE, Pa. (AP) \u2014 Police say a Pennsylvania man apply-
ing to become a state trooper has been arrested after saying
during a polygraph pretest that he had sex with an underage gi-
rl four years ago... She is now 19. Police say they charged
White on Thursday with four counts of unlawful sexual contact
with a minor and 10 counts of corruption of minors after inter-
viewing the woman and corroborating that information.
```

Passage [4]:

```
and he doesn't have a listed phone number...
```

*Summary:*

A 29-year-old Pennsylvania man, Joseph Adam White, was arrested after admitting to having sex with an underage girl during a polygraph pretest for a state trooper position [1][2][3]. The incident occurred at the Meadville barracks on January 15, where White was taking the police cadet lie detector exam [3]. During the pretest, White allegedly told the examiner that he had consensual sex and other contact with the girl in 2011, when she was underage [1][2][3]. The girl is now 19 years old [2][3]. Following an investigation and an interview with the woman, White was charged with four counts of unlawful sexual contact with a minor and 10 counts of corruption of minors [2][3]. As a result, White's application to become a state trooper was rejected, and he is currently facing charges [1][2][3].

*Ground truth:*

2013 A Pennsylvania man is behind bars after allegedly telling state police he'd had sex with an underage girl. What's unusual: He told them while applying for a job as a state trooper, the AP reports via the Patriot-News. Police say Joseph White, 29, made the admission during a polygraph pretest at the Meadville state police barracks on Jan. 15. He allegedly admitted to consensual sex and other sexual contact with the girl four years ago; she's now 19. Police say the victim later corroborated the incidents, which occurred in a field in South Shenango Township, the Meadville Tribune reports. White was arrested Thursday on multiple charges of unlawful sexual contact with a minor and corruption of minors, and jailed on $25,000 bond. Needless to say, he won't be getting an interview, quips the Patriot-News.

Figure 25: Llama-3.1-70B-Instruct in MultiNews

**GPT-4o in LoCoMo**

Passage [15] :

```
...\nCaroline said, \"Painting's great for expressing myself. I
love creating art! Here's a recent self-portrait I made last we-
ek. \" and shared a photo of a painting of a woman with a blue
face. \nMelanie said,\"Carolin ...
```

*Question:*
When did Caroline draw a self-portrait?

*Answer:*
Caroline drew a self-portrait last week [15].

*Ground truth:*
The week before 23 August 2023

Figure 26: GPT-4o in LoCoMo

**Claude-3.5-Sonnet in LoCoMo**

Passage [14] :

```
... I'm so stoked about it! Check it out!\" and shared a photo-
graphy of a man with purple hair and glasses taking a selfie.\n
Joanna said, \"Wow, your new hair color looks ...
```

*Question:*
What color did Nate choose for his hair?

*Answer:*
Nate dyed his hair purple, as revealed in a selfie he shared with Joanna [14]. He chose this bright and bold color because he wanted to stand out from regular options, reflecting his personality [14].

*Ground truth:*
purple

Figure 27: Claude-3.5-Sonnet in LoCoMo

**Phi-3.5-mini-Instruct in LoCoMo**

Passage [62] :

```
...  dogs love running around the park. But one thing they hate
is snow. I took them to a snowy one last winter and they were so
confused! They definitely prefer nice, sunny days in the grass.\
"and shared a photo of a dog wearing a shirt standing on a ...
```

*Question:*
How do Audrey's dogs react to snow?

*Answer:*
Audrey's dogs seem to be confused and not particularly fond of snow, as they had a difficult time adjusting during a snowy day visit to the park (Conversation [62]). Reference: Conversation [62]

*Ground truth:*
Confused

Figure 28: Phi-3.5-mini-Instruct in LoCoMo

---

**Llama-3.1-8B-Instruct in LoCoMo**

Passage [15] :

```
...   He's even keeping my other pets active.\"\nJoanna said, \"
Pets sure do make life better! Glad Max is bringing you lots of
joy.\"\n Nate said, \"Yep, totally! Pets make us so much happier
and never let us down. Have you ...
```

*Question:*
Who was the new addition to Nate's family in May 2022?

*Answer:*
Nate adopted a dog named Max in May 2022 [15].

*Ground truth:*
Max

Figure 29: Llama-3.1-8B-Instruct in LoCoMo

---

**Qwen2-57B-A14B-Instruct in LoCoMo**

Passage [15] :

```
...   Hey Mel, great to chat with you again! So much has happened
since we last spoke – I went to an LGBTQ conference two days ago
and it was really special. I got the chance to meet and ...
```

*Question:*
What workshop did Caroline attend recently?

*Answer:*
Caroline attended an LGBTQ conference two days ago [15].

*Ground truth:*
LGBTQ+ counseling workshop

Figure 30: Qwen2-57B-A14B-Instruct in LoCoMo

---

**Llama-3.1-70B-Instruct in LoCoMo**

Passage [13] :

```
... Nate said, \"Coconut milk ice cream is at the top of my list.
It's so smooth and creamy with a tropical coconut twist. Plus,
it's dairy-free for people who can't have lactose or who  ...
```

*Question:*
What is Nate's favorite dish from the cooking show he hosted?

*Answer:*
Nate's favorite dish from the cooking show he hosted is coconut milk ice cream [13].

*Ground truth:*
Coconut milk ice cream

Figure 31: Llama-3.1-70B-Instruct in LoCoMo

---

**GPT-4o in NIAH**

Passage [1] :

```
... Till recently I thought it didn't, but the truth turns out
to be  more complicated. Originally, yes, it was meaningless.
Now it seems to have acquired a meaning.  And yet those who di-
slike the term are probably right, because if it means what I
think it does, we don't need it The best way to start your day
in New York is to watch the sunrise from the Brooklyn Bridge
and have a bagel. ...
```

*Question:*
What is the best way to start your day in New York?

*Answer:*
The best way to start your day in New York is to watch the sunrise from the Brooklyn Bridge and then have a bagel [1].

*Ground truth:*
watch the sunrise from the Brooklyn Bridge and have a bagel

Figure 32: GPT-4o in NIAH

---

**Claude-3.5-Sonnet in NIAH**

Passage [4] :

```
... In Paris, the most enjoyable activity is strolling along the
Seine River and savoring a croissant.  For all practical purpos-
es, ...
```

*Question:*
What is the most enjoyable activity in Paris?

*Answer:*
According to the passage, the most enjoyable activity in Paris is strolling along the Seine River and savoring a croissant [4].

*Ground truth:*
strolling along the Seine River and savoring a croissant

Figure 33: Claude-3.5-Sonnet in NIAH

---

**Phi-3.5-mini-Instruct in NIAH**

Passage [1] :

```
... In Bangkok, the top thing to do is to visit the Grand Palace
and sample street food.. A top-ranked professional chess ...
```

*Question:*
What is the top thing to do in Bangkok?

*Answer:*
Visit the Grand Palace and sample street food [1].

*Ground truth:*
visit the Grand Palace and sample street food

Figure 34: Phi-3.5-mini-Instruct in NIAH

---

**Llama-3.1-8B-Instruct in NIAH**

Passage [1] :

```
... At the extreme end of the spectrum are crack and meth The
best thing to do in Beijing is to take a walk in Chaoyang Park
and have a cup of Espresso in the evenin ...
```

*Question:*
What is the best thing to do in Beijing ?

*Answer:*
The best thing to do in Beijing is to take a walk in Chaoyang Park and have a cup of Espresso in the evening [1].

*Ground truth:*
take a walk in Chaoyang Park and have a cup of Espresso in the evening

---

Figure 35: Llama-3.1-8B-Instruct in NIAH

---

**Qwen2-57B-A14B-Instruct in NIAH**

Passage [10] :

```
... the must-do is to explore Shinjuku Gyoen National Garden and
taste matcha tea
```

*Question:*
What is the must-do in Tokyo?

*Answer:*
The must-do in Tokyo is to explore Shinjuku Gyoen National Garden and taste matcha tea [10].

*Ground truth:*
explore Shinjuku Gyoen National Garden and taste matcha tea

---

Figure 36: Qwen2-57B-A14B-Instruct in NIAH

---

**Llama-3.1-70B-Instruct in NIAH**

Passage [26] :

```
... thanks to the internet, there The most effective way to lose
weight is by eating a balanced diet and exercising regularly.'s
a path. Anyon...
```

*Question:*
What is the most effective way to lose weight?

*Answer:*
The most effective way to lose weight is by eating a balanced diet and exercising regularly [26].

*Ground truth:*
by eating a balanced diet and exercising regularly.

---

Figure 37: Llama-3.1-70B-Instruct in NIAH

GPT-4o in Counting Stars

Passage [97] :

```
The little penguin counted 36 \u2605\n not speed up.  Essays sh-
ould do the opposite.  Essays should aim for maximum surprise.I
was afraid of flying for a long time and could only travel vica-
riously...
```

Passage [193] :

```
Socially they \nThe little penguin counted 76 \u2605\n'd seem
more like rants posted by nutjobs on their GeoCities sites than
the genteel and beautifully typeset compositions published in
The New Yorker...
```

*Question:*
Please help the little penguin collect the number of ⋆.

*Answer:*
{"little_penguin": [36, 76], "passage_id": [97, 193]}

36, 76

Figure 38: GPT-4o in Counting Stars

Claude-3.5-Sonnet in Counting Stars

Passage [18] :

```
Or that \nThe little penguin counted 58 \u2605\n's what it feels
like. What's really going on is that you have different expectat
...
```

Passage [36] :

```
but don't put bars on the windows. \nThe little penguin counted
106 \u2605\n[2] Even now I sometimes write essays that are not
meant ...
```

*Question:*
Please help the little penguin collect the number of ⋆.

*Answer:*
{"little_penguin": [58, 106], "passage_id": [18, 36]}

58, 106

Figure 39: Claude-3.5-Sonnet in Counting Stars

Phi-3.5-mini-Instruct in Counting Stars

Passage [18] :

```
... which is the reason they   began blogging in the first place
\nThe little penguin counted 16 \u2605\n....
```

Passage [114] :

```
... startup hubs is the degree to which people help one another
out \nThe little penguin counted 34 ...
```

*Question:*
Please help the little penguin collect the number of ⋆.

*Answer:*
{"little_penguin": [58, 106], "passage_id": [18, 36]}

16, 106

Figure 40: Phi-3.5-mini-Instruct in Counting Stars

---

**Llama-3.1-8B-Instruct in Counting Stars**

Passage [1] :

```
July 2010What hard liquor \nThe little penguin counted 65 \u2605
\n, cigarettes, heroin, and crack have....
```

Passage [58] :

```
... We would end up getting all the users \nThe little penguin
counted 58  ...
```

*Question:*
Please help the little penguin collect the number of ⋆.

*Answer:*
{"little_penguin": [65, 58], "passage_id": [1, 58]}

65, 58

Figure 41: Llama-3.1-8B-Instruct in Counting Stars

---

**Qwen2-57B-A14B-Instruct in Counting Stars**

Passage [1] :

```
July 2010What hard liquor \nThe little penguin counted 65 \u2605
\n, cigarettes, heroin, and crack have....
```

Passage [58] :

```
... We would end up getting all the users \nThe little penguin
counted 58  ...
```

*Question:*
Please help the little penguin collect the number of ⋆.

*Answer:*
{"little_penguin": [65, 58], "passage_id": [1, 58]}

65, 58

Figure 42: Qwen2-57B-A14B-Instruct in Counting Stars

---

**Llama-3.1-70B-Instruct in Counting Stars**

Passage [18] :

```
... Or that \nThe little penguin counted 58 \u2605\n's what it
feels like ....
```

Passage [36] :

```
but don't put bars on the windows. \nThe little penguin counted
106 \u2605\n[2] Even now I sometimes write  ...
```

*Question:*
Please help the little penguin collect the number of ⋆.

*Answer:*
{"little_penguin": [58, 106], "passage_id": [18, 36]}

58, 106

Figure 43: Llama-3.1-70B-Instruct in Counting Stars

