# OpenReview forum: "L-CiteEval: Do Long-Context Models Truly Leverage Context for Responding?"
_ICLR.cc/2025/Conference — ICLR 2025 Conference Withdrawn Submission_

### Official Review · Reviewer_69QF · 2024-10-29

**Soundness:** 2
**Presentation:** 3
**Contribution:** 2
**Rating:** 5
**Confidence:** 4

**Summary:**

The authors introduce the L-CiteEval benchmark to evaluate LLMs on long-context tasks with citations. L-CiteEval includes both real-world and synthetic tasks across 11 domains, with context lengths ranging from 8K to 48K tokens. Unlike existing benchmarks that often limit scope or rely on LLM judges, L-CiteEval assesses models based on both response and citation quality, using an automated evaluation suite for consistency and reproducibility. The evaluation reveals that open-source models tend to rely more on intrinsic knowledge, whereas closed-source models excel in citation accuracy. However, the authors note that current citation quality metrics may not fully capture nuanced aspects of citation faithfulness.

**Strengths:**

L-CiteEval addresses an important challenge in LLM evaluation. With extremely lengthy contexts, it is increasingly difficult for humans to verify model performance manually.

The use of automated evaluation over LLM-based judgments mitigates the biases that can be introduced by using models like GPT-4 as evaluators.

While previous work, such as Bai et al. (2024)’s LongCite, already introduced citation benchmarking, L-CiteEval’s approach adds automated quality checks on both response content and citation integrity.

**Weaknesses:**

The authors mention using padding to extend short-context datasets. Relying solely on Named Entity Recognition (NER) to filter padding data might not fully eliminate unwanted noise, potentially impacting model response consistency.

L-CiteEval assembles its benchmark from pre-existing datasets, applying a padding strategy to extend short-context data to the target lengths. This approach, while practical, does not introduce new data or methodologies for long-context representation, which limits the novelty of the contribution. The primary innovation seems to be the padding process.

As the authors acknowledge, the current citation metrics used in L-CiteEval may not fully capture the intricacies of citation faithfulness, particularly in complex, multi-reference scenarios. This limitation restricts the benchmark’s ability to comprehensively assess whether models are citing accurately and contextually, which is essential in long-context tasks.

**Questions:**

Have the authors compared model performance on genuinely long texts versus artificially padded ones? Insights into how padding affects the model’s citation accuracy and response relevance could inform whether padding is a viable approach for long-context benchmarks.

Did the authors consider alternative or additional metrics that might capture nuanced citation quality more effectively? For example, integrating metrics that evaluate citation relevance based on context similarity could provide a more robust evaluation of citation faithfulness.

---

### Official Review · Reviewer_HrPY · 2024-11-04

**Soundness:** 2
**Presentation:** 2
**Contribution:** 1
**Rating:** 1
**Confidence:** 5

**Summary:**

This paper introduces L-CiteEval, a multi-task benchmark for long-context understanding with citations for LLMs. Although the paper is well-written, it is potentially unfit for ICLR and best suited for linguistics venues.

**Strengths:**

1.) This paper introduces L-CiteEval, a benchmark for long-context understanding with citations for several open-source and closed-source LLMs.

2.) The paper is well-written in a particular context and a due literature review has been conducted.

3.) The analysis done in the paper is good.

**Weaknesses:**

1.) I found this paper unfit for ICLR. Though it is about datasets and benchmarks yet there is nothing about learning representations or core machine learning. The paper is best suited for core NLP venues. As we can go through the references section, almost every referenced paper is from arXiv, and the very few remaining ones are from computational linguistics.

2.) The x-axis in Fig.4(a) for all the subplots is wrong. It should not be (Easy, Medium and Hard, i.e. difficulty) but should be the values of varying context length. It is imperative that the authors double-check all axis labels and legends to ensure consistency with the described experiments and accurately reflect the data being presented.

3.) The legends on both the x-axis and the y-axis are missing in Fig.3 and Fig.4. Adding clear and descriptive axis labels to all figures to improve readability and interpretation of the results is a general practice that is followed in the scientific domain.

4.) The captions for all the figures should be descriptive enough instead of small one-liners. Detailed captions ensure that the figure conveys the requisite information and the reader is not compelled to read the complete main text.

5.) The claim “However, the closed-source LCM (GPT-4o) maintains a relatively stable performance, showing minimal degradation.” in line 418 is incorrect. It is clearly visible from several subplots in Fig.4(a) that there is high degradation. E.g., NarrativeQA F_1, GovReport F_1, NarrativeQA Rec., HotpotQA Rec., GovReport Rec. Moreover, just taking three points on the x-axis is not sufficient to establish a trend and thus claim a phenomenon. Since it is a benchmarking paper, it should be taken into account carefully. So, authors need to provide a more nuanced discussion of performance variations across different tasks and metrics, do extensive and accurate benchmarking, and thus derive valid inference and claims.

6.) In line 477, it should be “performance of retrieval”.

There may be other weaknesses too but I did not give the paper a second read.

**Questions:**

NA

---

### Official Review · Reviewer_YdsP · 2024-11-06

**Soundness:** 1
**Presentation:** 3
**Contribution:** 2
**Rating:** 5
**Confidence:** 3

**Summary:**

This paper introduces L-CiteEval, a benchmark specifically designed to evaluate long-context models (LCMs) on their ability to effectively use extended contexts for generating responses and maintaining citation faithfulness. With the rising use of LCMs for tasks that require processing extensive information, such as summarization and question answering over large documents, accurately assessing whether these models rely on the provided context or default to intrinsic knowledge is essential. L-CiteEval aims to address this need by providing a systematic, automated evaluation suite across 11 diverse tasks, with context lengths ranging from 8K to 48K tokens. This setup enables a robust assessment of how well LCMs handle the unique demands of long-context scenarios.

L-CiteEval evaluates models along two dimensions: response quality, which measures the relevance and accuracy of the generated answers, and citation faithfulness, which assesses whether the model’s responses are appropriately grounded in the source material. To explore the impact of context length and task complexity, the benchmark includes two variants—L-CiteEval-Length and L-CiteEval-Hardness—allowing for an analysis of model performance based on different context lengths and task difficulties. Through these variants, L-CiteEval provides a nuanced view of how well models manage extended contexts and remain faithful to provided information, which is particularly relevant for applications in research and academia where accurate source attribution is critical.

The authors benchmarked 11 long-context models, including both open-source and closed-source options, revealing significant differences in citation faithfulness. Their findings indicate that open-source models often struggle to maintain citation accuracy, with a tendency to rely on intrinsic knowledge rather than contextual information. Notably, incorporating the Retrieval-Augmented Generation (RAG) approach improved citation accuracy but slightly reduced the quality of response generation. Additionally, the study identifies correlations between a model’s citation performance and its attention mechanisms, offering insights into how model architecture may influence citation faithfulness.

While the paper makes a contribution by introducing a comprehensive benchmark, it does not propose new technical methods or model architectures. Additionally, it remains unclear how the insights gained from its evaluations will directly inform or advance the development of LCMs. Although L-CiteEval addresses a gap in assessing LCMs’ contextual understanding and citation accuracy, the paper could benefit from more explicit guidance on how these findings could be leveraged to improve LCM design and performance in practical applications.

**Strengths:**

1) L-CiteEval thoroughly assessed LCMs on response quality and citation faithfulness, covering diverse tasks with up to 48K tokens, filling a key gap in LCM evaluation.
2) The benchmark’s emphasis on citation accuracy is crucial for research applications, ensuring models rely on provided context rather than intrinsic knowledge.
3) Evaluating both open-source and closed-source models, the paper identifies performance gaps, highlighting challenges for open-source LCMs and the benefits of Retrieval-Augmented Generation (RAG) for citation reliability.

**Weaknesses:**

1) The paper identifies gaps in citation accuracy but explores few methods beyond RAG to improve performance.
2) Heavy dependence on automated scoring may overlook nuances; human evaluation could add reliability.
3) It is unclear how the insights gained from its evaluations will directly inform or advance the development of LCM in practice.

**Questions:**

1) Are certain tasks particularly challenging for LCMs in terms of citation faithfulness?
2) Did the authors observe any trends in model performance as context length increased? Are there critical thresholds beyond which models struggle with citation or response quality?

---

### Official Review · Reviewer_MN48 · 2024-11-10

**Soundness:** 2
**Presentation:** 3
**Contribution:** 1
**Rating:** 3
**Confidence:** 3

**Summary:**

The authors introduce a multi-task benchmark called L-CiteEval to evaluate long-context models (LCM) from the perspective of generation quality and citation quality (i.e., how the response is correctly supported by information from the context). Compared to existing long-context evaluation benchmarks, L-CiteEval has a wider range of tasks, data points with longer context lengths, and strictly categorizes tasks by length intervals. The authors also curate two variants, L-CiteEval-Length and L-CiteEval-Hardness, to decouple the difficulty of the tasks from the long-context understanding in the evaluation. The authors test 11 LCMs on L-CiteEval and find that 1) open-source models underperform closed-source models in terms of citation quality; 2) RAG improves LCMs' faithfulness; 3) There is a correlation between the model’s citation generation process and its attention mechanism.

**Strengths:**

1. The proposed benchmark contains more tasks and longer contexts that can better evaluate LCMs.

2. The authors conduct comprehensive experiments on an extensive collection of LCMs.

**Weaknesses:**

1. Besides more tasks, longer contexts, and controlable lengths, the introduced benchmark L-CiteEval has limited novelty compared to the existing benchmark LongCite. The authors fail to demonstrate the necessity of L-CiteEval - what are the research questions that can be answered only by L-CiteEval but not LongCite.

2. In the related works, the authors mention that "L-CiteEval relies entirely on automatic evaluation metrics without reliance on GPT-4 or human judgments" and list it as an advantage of L-CiteEval compared to LongCite. But in the limitation section (Section A), the authors state "current evaluation metrics still heavily rely on human judgment or model outputs." as the limitation of the work. This is confusing.

3. There exist errors in the manuscript. For example, the x-axis of Figure 4(a) is inconsistent with its caption and its reference section (Section 4.2.1). This causes miscomprehension of the section and the figure. A typo on line 253, L-CiteEval-Quality -> L-CiteEval-Hardness.

**Questions:**

1. Are the evaluation metrics entirely automatic or not? How are the precision and recall evaluated on the QA benchmarks for the quality evaluation?

---

### Note · Authors · 2024-12-13

I have read and agree with the venue's withdrawal policy on behalf of myself and my co-authors.